# Whole-genome sequencing of *Schistosoma mansoni* reveals extensive diversity with limited selection despite mass drug administration

Duncan J. Berger [1,2,9 ✉], Thomas Crellen [1,3,4,9], Poppy H. L. Lamberton [3,5], Fiona Allan[6], Alan Tracey [1], Jennifer D. Noonan [7], Narcis B. Kabatereine[8], Edridah M. Tukahebwa[8], Moses Adriko[8], Nancy Holroyd[1], Joanne P. Webster [2,3,10 ✉], Matthew Berriman [1,10 ✉] & James A. Cotton [1,10 ✉]

Control and elimination of the parasitic disease schistosomiasis relies on mass administration of praziquantel. Whilst these programmes reduce infection prevalence and intensity, their impact on parasite transmission and evolution is poorly understood. Here we examine the genomic impact of repeated mass drug administration on *Schistosoma mansoni* populations with documented reduced praziquantel efficacy. We sequenced whole-genomes of 198 *S. mansoni* larvae from 34 Ugandan children from regions with contrasting praziquantel exposure. Parasites infecting children from Lake Victoria, a transmission hotspot, form a diverse panmictic population. A single round of treatment did not reduce this diversity with no apparent population contraction caused by long-term praziquantel use. We find evidence of positive selection acting on members of gene families previously implicated in praziquantel action, but detect no high frequency functionally impactful variants. As efforts to eliminate schistosomiasis intensify, our study provides a foundation for genomic surveillance of this major human parasite.

[1] Wellcome Sanger Institute, Wellcome Genome Campus, Hinxton, Cambridgeshire, UK. [2] Department of Pathology and Pathogen Biology, Centre for Emerging, Endemic and Exotic Diseases, Royal Veterinary College, University of London, Herts, UK. [3] Imperial College London, Department of Infectious Disease Epidemiology, London, UK. [4] Big Data Institute, Nuffield Department of Medicine, University of Oxford, Oxford, UK. [5] Institute for Biodiversity, Animal Health, and Comparative Medicine, and Wellcome Centre for Integrative Parasitology, University of Glasgow, Glasgow, UK. [6] The Natural History Museum, Department of Life Sciences, London, UK. [7] Institute of Parasitology, Faculty of Agricultural and Environmental Sciences, McGill University, Montreal, Quebec, Canada. [8] Vector Borne & Neglected Tropical Disease Control Division, Ministry of Health, Kampala, Uganda. [9] These authors contributed equally: Duncan J. Berger, Thomas Crellen. [10] These authors jointly supervised this work: Joanne P. Webster, Matthew Berriman, James A. Cotton. ✉email: db22@sanger.ac.uk; jowebster@rvc.ac.uk; mb4@sanger.ac.uk; jc17@sanger.ac.uk

Schistosomiasis is one of the most important neglected tropical diseases (NTDs), currently infecting over 240 million people across 78 countries, primarily in sub-Saharan Africa[1]. In 2001 the World Health Organization (WHO) endorsed preventive chemotherapy, in the form of mass drug administration (MDA) of praziquantel, as the primary strategy for schistosomiasis control[2]. Praziquantel MDA aims to provide periodic, typically annual, treatment of large populations at risk of disease, focusing primarily on school-aged children, a group shown to have the highest infection prevalence, intensity and future morbidity risks[3,4]. MDA in sub-Saharan Africa commenced in 2003, starting in Uganda, and in 2019 alone, MDA programmes delivered praziquantel to 105.4 million people[5].

These programmes have, in general, been effective at reducing the prevalence and intensities of infections in targeted regions[6–8]. Following these early successes, in 2012, the WHO set goals of controlling schistosomiasis morbidity (defined as reducing the prevalence of heavy intensity infections to <5% across sentinel sites) by 2020, eliminating schistosomiasis as a public health problem (EPHP, defined as reduced the prevalence of high-intensity infections to <1% across all sentinel sites) by 2025 and in selected regions, complete interruption of transmission, by 2025[8,9]. Most recently, the revised WHO 2021–2030 NTD roadmap was launched, with its more ambitious goal to achieve EPHP in all 78 endemic countries, and interruption of transmission in selected regions, by 2030[10].

However, despite the many successes to date, numerous persistent schistosomiasis hotspots remain[8,11–14]. The precise reasons for which are largely unknown but they present substantial challenges toward reaching these ambitious 2030 targets. Accurately measuring changes in parasite populations will be a key part of understanding the current and future impact of MDA[15,16]. Populations of *S. mansoni* appear, in general, to be structured at national or continental scales, with several million nucleotide variants differentiating populations in East and West Africa[17,18]. However, population genetic studies at smaller scales have reported panmictic populations with high within-host variability[19–21]. Estimation of key population parameters such as genetic diversity, population structure and parasite relatedness can all be used to infer recent demographic changes and the efficacy of interventions. While a number of population genetic studies have assessed how *Schistosoma* spp. populations have changed following MDA[22–24], these have been restricted to a limited number of molecular markers. By contrast, few studies have analysed whole-genome or exome sequence data from populations of *S. mansoni*[17,25] or other schistosomes[17,25–27].

These genomic approaches have the capacity to resolve parasite population structure, detect localised signatures of positive selection and inform epidemiological models of schistosomiasis[28]. For example, a reduction in the genetic diversity of a population might suggest the occurrence of a population bottleneck followed by inbreeding and the accumulation of deleterious alleles, which could be indicative of a population close to elimination[29]. Evidence of localised selection in the genome or the emergence and rapid rise in frequency of novel genotypes—particularly in the absence of a bottleneck, could indicate that a population is rapidly evolving in response to control measures.

The degree to which parasite population structure is influenced by current control measures is unknown, but adaptation is a particular concern. MDA programmes are highly dependent on the continued efficacy of praziquantel, the only drug available for widespread use that is effective against all human-infective schistosome species[30–32]. Drug resistance is an increasingly common consequence of large-scale treatment regimes and has emerged in veterinary systems to all major classes of anthelmintics[33–35]. Whilst there is no consensus as to its precise genetic basis, resistance to praziquantel can be experimentally induced in laboratory populations of *S. mansoni*[36–39]. Furthermore, several studies have reported potential reduced anthelmintic efficacy in endemic regions[40–42], although suggestions of widespread praziquantel resistance remain controversial[43–47]. In programmes that target only school-age children or other subsets of the population, there will be large refugia from drug selection[48]. However, with the proposed expansion of human MDA programmes to wider community-level coverage, praziquantel use is expected to exert increasing selective pressure for reduced praziquantel susceptibility. Surveillance for reduced praziquantel efficacy amongst MDA programmes currently relies only on slide-based egg counts before and after treatment to estimate egg reduction rates (ERR); the proportional reduction in egg output following treatment[49–51]. Genome-wide approaches have, in contrast, the potential to detect and monitor markers to track the spread of resistance through natural populations[16].

At the beginning of the first national MDA programme in Uganda in 2003, a 2-year school-based longitudinal survey in a high transmission region found evidence of natural praziquantel tolerance in these *S. mansoni* populations[20]. A survey involving the same schools in 2014 found that the ERR was significantly lower in parasites from children in schools that had received 8–9 rounds of annual MDA compared with schools that had received 1–5 previous rounds[42]. Together these studies suggested that the selective pressure of repeated MDA has acted on pre-existing genetic variation in praziquantel susceptibility resulting in reduced praziquantel efficacy.

In order to elucidate the genomic impact of long-term MDA and gain insights into the genomic diversity and potential gene flow amongst natural *S. mansoni* populations, we performed whole-genome sequencing on parasites collected during the 2014 survey[42]. We sequenced individual miracidia from children attending primary school in regions of either long-term MDA pressure (8–9 previous annual rounds) or short-term MDA pressure (1 previous round) from two Ugandan districts ~100 km apart. These children exhibited a range of parasite clearance phenotypes following praziquantel treatment, of which low clearance is potentially indicative of praziquantel-resistant parasites[42]. Using these data we report the genetic structure of *S. mansoni* across these two districts and determine whether there are observable recent changes in schistosome diversity that could be attributed to repeated praziquantel treatment. Based on data from pre- and post-treatment samples and the treatment response of individual children we investigated whether a single round of praziquantel treatment could reduce host infrapopulation diversity. Finally, we looked for evidence of recent selection which might explain the apparently reduced praziquantel efficacy in regions of long-term MDA and whether any variants are strongly associated with drug response. This study represents the largest whole-genome sequencing study, and the first whole-genome analysis of the effect of drug treatment, on schistosome populations from endemic regions.

## Results

**Whole-genome sequencing of individual miracidia.** We selected 222 larval schistosomes (miracidia) sampled in 2014 from children attending four primary schools in Eastern Uganda (Table 1, Fig. 1)[42] for whole-genome sequencing. The majority of miracidia were from Mayuge district ($n = 181$; from 31 children), where MDA had occurred annually in primary schools since 2003 (with the exception of 2008–2009). All three of the study sites in Mayuge (Bugoto, Bwondha and Musubi schools) were within 0.5 km of the shore of Lake Victoria. The remaining miracidia were sampled from Kocoge school ($n = 17$; from 3 children),

**Table 1 Sample information and history of praziquantel exposure.**

| District | School | MDA exposure | Mean ERR (Model 95% BCI) | Miracidia sequenced | Passed quality control | Children |
|---|---|---|---|---|---|---|
| Mayuge (Lake Victoria) | Bugoto | High (9 rounds) | 89.63 (77.65–96.65) | 86 | 75 | 12 |
| Mayuge (Lake Victoria) | Bwondha | High (9 rounds) | 92.10 (87.42–95.71) | 67 | 60 | 10 |
| Mayuge (Lake Victoria) | Musubi | High (8 rounds) | 91.98 (82.35–97.69) | 51 | 46 | 9 |
| Mayuge (Lake Victoria) | Kocoge | Low (1 round) | 98.87 (98.07–99.60) | 18 | 17 | 3 |

Schools within districts were assigned a mass drug administration (MDA) exposure category based on the number of previous annual rounds of treatment. A total of 222 miracidia were sequenced and 198 passed quality control measures from a total of 34 hosts unevenly distributed across schools and districts. Egg reduction rate (ERR) values for each school population are taken from Crellen et al.[42] and are given as the mean ERR value and 95% Bayesian credible interval (BCI).

located in Tororo district, which is close to the Kenyan border and ~100 km from Mayuge district. MDA commenced in Tororo district from 2013, indicating one previous round of treatment prior to our sampling. We also included *S. mansoni* sequence data from Crellen et al.[17] of adult worms isolated from Senegal, Cameroon, Kenya, Puerto Rico, Guadeloupe (3 samples) and two samples from Uganda from Walukuba school (Lake Albert) and Buloosi school (Lake Victoria). Alignment of miracidia sequence reads to the *S. mansoni* genome revealed high per-sample mapping rates (median 88.78% of reads mapped with mapping quality MQ >40; Supplementary Data 1). Reads mapped to the majority of the genome (median of 97.83% of bases covered across all samples), although the median read depth was highly variable across samples (0.13–58x) and substantially lower over the Z-specific region of the Z chromosome due to the inclusion of both male (ZZ) and female (ZW) miracidia in our sample set (Supplementary Data 1, Supplementary Fig. 1). Across all 207 samples, we produced a high-quality variant dataset (Supplementary Fig. 2) comprising 6,268,157 single-nucleotide polymorphisms (SNPs) and 442,787 indels (including mixed SNP/indel variant sites) from 198 miracidia and 9 adult worms.

**Population structure of Ugandan *Schistosoma mansoni*.** We characterised the population structure at different spatial scales using a subset of 1,344,965 autosomal variants, filtered to remove variants in strong linkage disequilibrium. Principal component analysis revealed separation of *S. mansoni* parasites by country along the first principal component and separation of the Ugandan samples by proximity to Lake Victoria along the second principal component (Fig. 2a). While the first four principal components explained 60% of the total variance, they did not differentiate parasite populations within Mayuge district by school or infrapopulation (Fig. 2a and b, Supplementary Figs. 3 and 4). Phylogenetic analysis showed that parasites from outside Uganda and the parasites from Tororo district formed distinct clades but there was no discernable structure within parasites from Mayuge district (Fig. 2c). ADMIXTURE analysis also consistently differentiated parasites of Tororo district from those of Mayuge, although cross-validation analysis (Supplementary Fig. 5) could not identify the most likely number of subpopulations (*K*), we confirmed this for a range of *K* values (Fig. 2f). Effective population size ($N_e$) showed broadly consistent population histories across all schools (Supplementary Figs. 6 and 7). Estimates of $N_e$ over time plateau ~10,000 years ago and were consistent between the Lake Victoria schools ($3.30–3.69 \times 10^4$) and slightly reduced in Kocoge school ($2.61 \times 10^4$).

Genome-wide estimates of median nucleotide diversity within parasite populations were consistent across all schools ($\pi_{Bugoto}$, $\pi_{Bwondha}$, $\pi_{Musubi}$ = 0.0033, $\pi_{Kocoge}$ = 0.0032; Fig. 2d). Estimates of relative ($F_{ST}$) and absolute ($d_{XY}$) differentiation showed low levels of differentiation but slightly elevated values between parasites sampled from schools in different districts (range $d_{XY}$ = 0.0034, range $F_{ST}$ = 0.0233–0.0247), compared with those sampled within the same district (range $d_{XY}$ = 0.0033, range $F_{ST}$ = 0; Fig. 2e).

Analysis of parasite populations within individual children showed approximately equivalent levels of nucleotide diversity (range $\pi$ = 0.0027–0.0036) to the range found within schools (Supplementary Fig. 8a). Differentiation between infrapopulations was generally low, both between schools (range $F_{ST}$ = 0–0.0339, range $d_{XY}$ = 0.0032–0.0038; Supplementary Data 2) and within schools (range $F_{ST}$ = 0–0.0123, range $d_{XY}$ = 0.0032–0.0037; Supplementary Data 2). There was no significant difference in nucleotide diversity between infrapopulations from male or female schoolchildren ($p$ = 0.2249, Supplementary Fig. 8b). Likewise, there was no significant relationship between nucleotide diversity

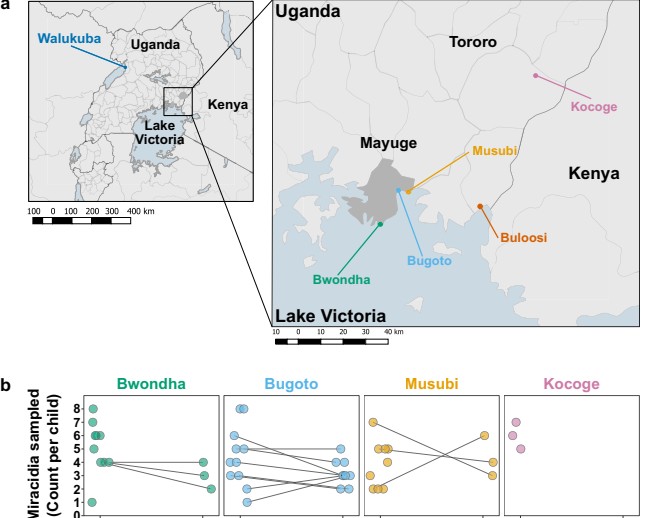

**Fig. 1 Study design and summary of miracidia stage samples sequenced.**
**a** Sampling was performed in two districts in Uganda; Mayuge district and
Tororo district. Within those regions four schools were selected, three from
Mayuge district which borders Lake Victoria: Bwondha (green), Bugoto
(light blue) and Musubi (yellow). One school from the Tororo district, close
to the Kenya border, was chosen: Kocoge (pink). Mayuge district and
Tororo district primary schools in each region have been under long-term
and short-term mass drug administration (MDA) pressure, respectively.
**b** Pre- and post-treatment sampling information. Circles indicate the
numbers of miracidia sampled at week 0 (immediately before treatment)
and 4 weeks post-treatment, lines indicate miracidia sampled at both time
points from the same children, only samples passing read sequence quality
control are included.

and child age ($R^2 = 0.051$, $p = 0.2$, Supplementary Fig. 8c),
infrapopulation size ($R^2 = 0.0099$, $p = 0.57$, Supplementary
Fig. 8d). Pairwise comparisons also revealed no evidence of first,
second or third order kinship relationships between any miracidia.
Analysis of differential coverage between pseudoautosomal and Z-
specific regions of the *S. mansoni* Z chromosome identified 103
miracidia samples as female and 95 as male (Supplementary Fig. 9,
Supplementary Data 3).

**Population size changes in regions of long-term MDA.** We
examined genome-wide allele frequency patterns to determine
whether there was any evidence of a recent reduction in popu-
lation size. Evidence of a recent population bottleneck could
indicate that school-based MDA coverage has been sufficiently
high to impact *S. mansoni* populations in Mayuge district despite
the refugia in untreated hosts, such as adults, pre-school-aged
children, non-school-attending children and snail vectors. Ana-
lysis of the one-dimensional site frequency spectra (1D-SFS) for
each school population showed an abundance of rare (singleton
and doubleton) alleles, suggestive of a large or growing popula-
tion (Fig. 3a, Supplementary Fig. 10)[52]. Genome-wide Tajima's D
estimates were negative in all populations suggesting an excess of
low-frequency polymorphisms compared to expectations (Fig. 3b,
Supplementary Fig. 11). A negative Tajima's D is consistent with
a population expansion after a selective sweep although genome-
wide Tajima's D and 1D-SFS results are broadly consistent
between the two districts, so it is unlikely this is due to MDA[53].

**Evidence of positive selection in regions of long-term MDA.**
Based on the differences in past treatment history with

praziquantel and genomic evidence of geographical separation
between populations, we screened for evidence of recent selection
within Mayuge district and between Mayuge and Tororo districts
using a number of genome-wide methods. Multiple regions were
identified by the integrated haplotype score (iHS) test as being
under strong selection in Mayuge where we expect selection for
survival following praziquantel treatment to be strongest (Fig. 4a).
Several of these regions also showed high genetic differentiation or
differences in haplotype frequencies between populations from
Mayuge and Tororo District using the fixation index ($F_{ST}$) and
cross-population extended haplotype homozygosity (XP-EHH)
statistics, respectively (Fig. 4c and d). We also determined whether
any of these selected regions intersected with regions of reduced
genetic diversity in the Mayuge samples (compared to Tororo)
which could indicate targeted selection on a locus (Fig. 4e).

Overall, we identified 25 non-redundant regions (range = 0.010–
1.37 Mb, total length = 7.86 Mb) with extreme iHS, $F_{ST}$ or XP-EHH
scores (Fig. 4, Supplementary Data 4 and 5) encompassing 183 genes
(representing 1.80% of all coding genes), of which 176 contained at
least one variant in one or more samples (Supplementary Data 6, 7).

The strongest signals of selection were detected within
chromosome 2. Our tests found evidence of geographically
localised selection with extreme iHS values in seven regions of
this chromosome (total length 2.69 Mb; Fig. 3a). Two of these
regions overlapped with elevated XP-EHH, $F_{ST}$ values and
reduced relative nucleotide diversity (Fig. 3c–e, Supplementary
Fig. 12, Supplementary Data 4). Within one of these regions we
identified a single locus (47.06–47.34 Mb) containing four
sodium/potassium/calcium exchanger proteins (Smp_170450,
Smp_336070, Smp_328710, Smp_094390; Supplementary Fig. 13,
Supplementary Data 6). Broad families of calcium ion transpor-
ters, including voltage-operated $Ca^{2+}$ channels, have been
suggested as a means of praziquantel mediated calcium
uptake[54–57] with changes in these channels being suggested as
potential means of resistance[58,59]. Of these four genes, three
contained variants corresponding to predicted missense, non-
sense and splicing variants (7 in Smp_094390, 19 in Smp_170450
and 1 in Smp_336070). All the non-synonymous variants were
found in Smp_170450 and had a MAF>10% within the Mayuge
parasite population (Supplementary Data 7).

Several other candidate regions were identified including a
region of extreme iHS scores in the Mayuge population at the
start of chromosome 4 that encompassed 11 genes (Supplemen-
tary Fig. 14). This region included a gene predicted to encode an
ATP-binding cassette (ABC) transporter *smdr1* (Smp_063000)
although the entire region had elevated relative nucleotide
diversity compared to Tororo and iHS scores were higher in
Tororo samples compared to Mayuge samples across the 56 kb
span of this gene.

Further evidence of positive selection was found at the start of
chromosome 2 (3.55–4.81 Mb, Supplementary Fig. 15). We also
found comparatively lower nucleotide diversity in this region
within the Mayuge populations compared to the Tororo
population. This region encompassed 19 genes all of which
contained one or more variants but no clear candidates for
praziquantel-linked selection were identified. Similar results were
found when other candidate regions were examined (Supple-
mentary Figs. 16 and 17). We also found no obvious evidence of
selection acting on the $Ca^{2+}$-permeable transient receptor
potential (TRP) channel (Smp_246790) recently identified as a
likely target of praziquantel activity and possible means of
praziquantel resistance (Supplementary Fig. 18)[60–62].

**Short-term impact of treatment on genomic diversity.** We
stratified populations based on host clearance phenotype and time of

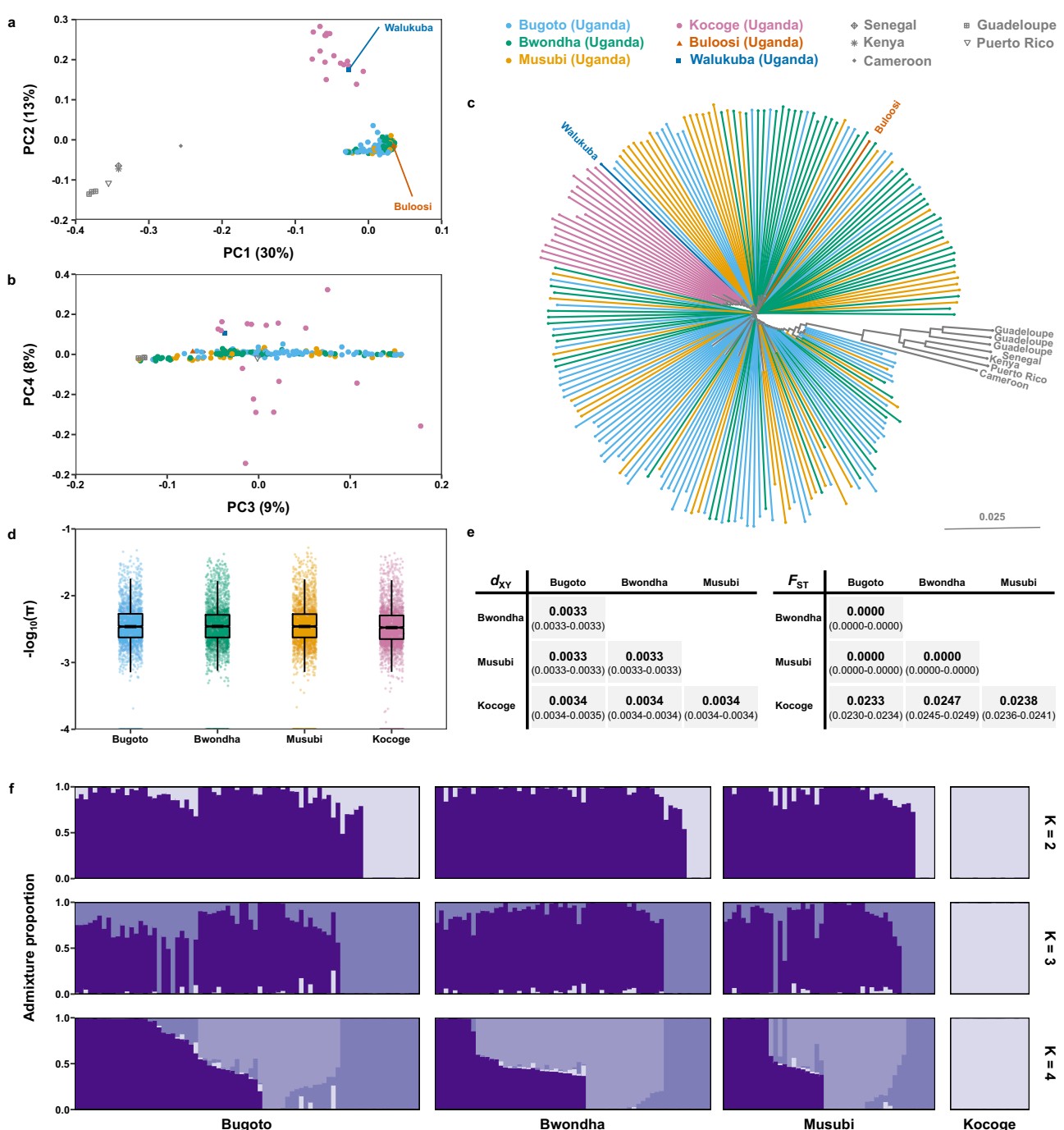

**Fig. 2 Schistosoma mansoni population structure.** Principal component analysis (PCA) of genetic differentiation within and between the 198 S. mansoni isolates produced in the present study, plus nine previously published samples from Uganda and elsewhere. **a** Principal components 1 and 2 and **b** components 3 and 4 with the first four principal components accounting for 60% of the total variance. Points represent samples from three schools Mayuge district (Uganda), Bugoto (light blue), Bwonda (green) and Musubi (yellow) and one school from Tororo district, Kocoge (pink). We included nine additional previously published samples, one sample was from Buloosi school (orange) ~40 km east of Mayuge district, a second sample was from Walukuba school (dark blue) near Lake Albert and the remaining samples (grey) were from Guadeloupe, Senegal, Kenya, Puerto Rico and Cameroon. **c** Midpoint rooted neighbour-joining phylogeny showing the relatedness between samples, branches are coloured based on the school or region where sample collection occurred. **d** Autosomal nucleotide diversity ($\pi$) values, calculated as the mean of non-overlapping 5 kb windows for each school population: Bugoto ($n = 75$ miracidia), Bwondha ($n = 60$), Musubi ($n = 46$) and Kocoge ($n = 17$). For all boxplots, the central line indicates the median, the top and bottom edges of the box indicate the 25th and 75th percentiles, respectively. The maximum whisker lengths are specified as 1.5 times the interquartile range. **e** Pairwise comparisons of fixation index ($F_{ST}$) and absolute divergence ($d_{XY}$) between each school population. Both statistics were calculated using autosomal variants in non-overlapping 5 kb windows using the same sample sizes as (**d**). Median values for each comparison are shown in bold, numbers in parentheses represent the 95% bootstrap confidence intervals around the median. **f** ADMIXTURE plots illustrating the population structure, assuming 2–4 populations are present ($K$), using 10-fold cross-validation and standard error estimation with 250 bootstraps. Y-axis values show admixture proportions for different values of $K$ ($K = 2$–4), each shade of purple indicates a different population.

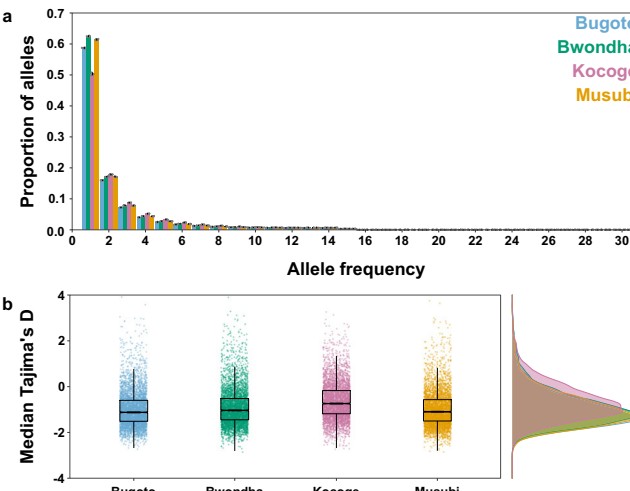

**Fig. 3 Genome-wide allele frequency patterns. a** One-dimensional site frequency spectra for each parasite population sampled from children in each school: Bugoto (light blue), Bwondha (green), Musubi (yellow) and Kocoge (pink). For each school, miracidial populations were subsampled ($n = 15$ miracidia per school) and site frequency spectra were calculated, this was repeated for a total of five replicates. The x-axis represents the derived allele frequency and y-axis represents the proportion of sites at each allele frequency. Coloured bars represent the median proportion of sites across all replicates for each school, black error bars represent the standard deviation around the median for all replicates, grey points represent the individual results for each replicate. **b** Median Tajima's D values calculated in 5 kb windows across each autosome for each school population. For all boxplots, the central line indicates the median, the top and bottom edges of the box indicate the 25th and 75th percentiles, respectively. The maximum whisker lengths are specified as 1.5 times the interquartile range.

sampling, forming three groups (Fig. 5). Samples from children with no miracidial hatching after treatment were classified as 'good clearers', samples from children with post-treatment miracidial hatching were grouped as either 'pre-treatment' if they were sampled prior to treatment or 'post-treatment' if they were sampled four weeks post-treatment. We excluded all 17 samples from Tororo district (Kocoge school) as all of these samples were derived from children with 'good' clearance phenotypes. Both pre- and post-treatment populations had slightly lower levels of nucleotide diversity (median $\pi_{\text{Pre-treatment}} = 0.0030$, median $\pi_{\text{Post-treatment}} = 0.0030$) compared to parasite populations from children with good clearance phenotypes (median $\pi_{\text{Good clearers}} = 0.0032$; Fig. 5a). Estimates of relative ($F_{\text{ST}}$) and absolute ($d_{\text{XY}}$) differentiation between these three populations showed minimal variation between any populations for both metrics (Fig. 5b), and in particular, showed no difference in allele frequencies between pre- and post-treatment samples from poor clearance individuals. Pairwise genome-wide $F_{\text{ST}}$ analysis did not highlight any regions of elevated values between any of the three populations (Fig. 5c).

**Genome-wide association for parasite clearance.** Given the importance of understanding the genetic mechanism of praziquantel resistance, we also formally tested for an association between reduced clearance phenotypes and autosomal *S. mansoni* variants. We conducted this analysis despite the limited sample size and number of children with poor clearance phenotypes (Table 1). First, we performed a binary trait association between miracidia sampled from children with good clearance phenotypes (where treatment appeared to be highly effective) and miracidia

isolated post-treatment from children with poor clearance phenotypes (where miracidia are potentially derived from parasites that survived treatment; Fig. 5d). Second, we performed quantitative trait genome-wide association analysis using estimated ERR as the phenotype (Fig. 5e). The binary association test found no variants significantly associated with a poor clearance phenotype. The quantitative trait association test only identified four widely distributed variants associated with lower estimated ERR (Fig. 5e, Supplementary Data 8) and none were predicted to alter protein sequences (2 were intergenic SNPs, 1 SNP was found in a 3′ UTR and the other variant was an intronic indel).

## Discussion

The newly launched WHO NTD roadmap has set clear goals for the control and elimination of schistosomiasis and states that this will require extending mass treatment to all endemic populations and age groups[10]. Meeting these goals will require a greater understanding of how current and future interventions impact schistosome populations. This includes population change in response to repeated MDA as well as monitoring for the potential emergence and spread of praziquantel resistance. We therefore performed whole-genome sequencing of individual *S. mansoni* miracidia collected from epidemiologically important, high prevalence settings[42]. We sampled parasites from two districts in Uganda with different histories of praziquantel MDA and corresponding reduced efficacy of praziquantel in schools with a higher exposure to MDA[42].

Our analyses, which included adult worms from multiple countries, revealed clear population structure over national and continental scales. Within Uganda, we could distinguish parasites from Lake Albert and Lake Victoria, ~350–450 km apart, as well as miracidial populations sampled from two districts in Eastern Uganda, ~100 km apart. Parasites collected from children in three schools bordering Lake Victoria formed a panmictic and highly diverse population, with equivalent levels of diversity found within populations sampled from individual children as from across the district. This is consistent with population genetic surveys that have found high genetic diversity but minimal population structure among schistosomes sampled from different shoreline regions around Lake Victoria[18–20,22,63,64]. We found no indication that this genomic diversity was correlated with host-specific factors, despite some prior population genetic studies suggesting variance by host age or sex[20,65–70]. Consistent with the limited number of miracidia sampled relative to the high within-host diversity, we found no related miracidia (up to 3rd degree). Estimates by Aemero et al.[71] found that the majority (66–92%) of adult worm pairs from hosts are unrelated although sibship reconstruction models have shown that accurate estimations of relatedness depend on the number of sampled parasites[72].

Parasite genetic diversity is thought to be influenced by a range of epidemiological and host-specific factors[73]. At the intermediate host level, individual snails are typically found to support and shed a limited number of parasite genotypes[74] and have short lifespans. However, a high proportion of adults in these shoreline communities work in the fishing industry which leads to families moving around the Lake, and possibly crossing national borders, to increase their fishing yield[75]. This mobility leads to individuals being exposed to, and infecting, different snail populations around the lake shores and so facilitates gene flow and high within-host diversity.

This large population and high level of diversity would be expected to provide a strong adaptive advantage to *S. mansoni* populations allowing praziquantel resistance to develop more rapidly compared to less diverse populations[76,77]. At the district level, we found negative genome-wide Tajima's D and an excess

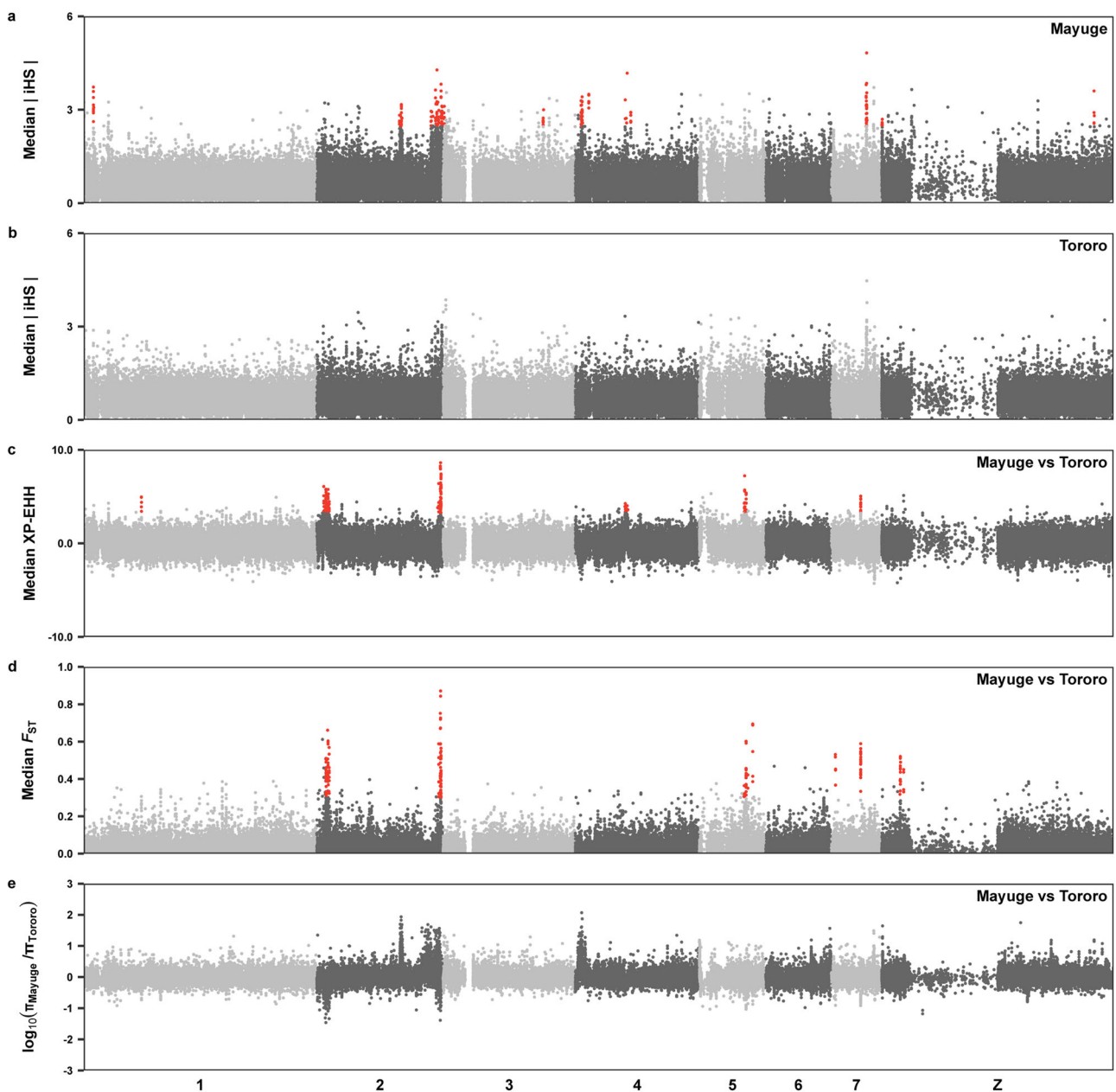

**Fig. 4 Signatures of recent selection. a** Genome-wide integrated haplotype scores (iHS) within the Mayuge population ($n = 181$) a region of long-term MDA pressure (8–9 previous annual rounds). **b** Genome-wide integrated haplotype scores (iHS) within the Tororo populations ($n = 17$), a region of short-term MDA pressure (1 previous round with limited coverage). Points represent median values of all variants in 2 kb non-overlapping windows along the eight *S. mansoni* chromosomes (shown in alternating shades of grey). **c** Genome-wide cross-population extended haplotype heterozygosity values (XP-EHH) between populations from Mayuge district and Tororo district, calculated as the median of all variants in 2 kb non-overlapping windows. **d** Fixation index ($F_{ST}$) values between populations from Mayuge district and Tororo district, calculated as the mean of $F_{ST}$ scores in non-overlapping 2 kb windows. **e** Nucleotide diversity was calculated as the mean of 2 kb windows across each chromosome for each district, the ratio between Mayuge and Tororo 2 kb windows is shown. Windows with the highest 0.25% of values in the Mayuge iHS, $F_{ST}$ and XP-EHH statistics (shown as red points) were used to define candidate regions of selection. Windows with a 300 kb of each other were joined into continuous regions of selection, any of those regions with <8 windows with elevated values across any statistic were discarded (Supplementary Data 4).

of rare alleles in both Mayuge and Tororo populations. This could indicate a recent population bottleneck, followed by a population expansion, but it is unclear whether this is due to MDA. The negative Tajima's D signal is similar in both Mayuge and Tororo populations, and is present genome-wide suggesting that it is not due to a selective sweep at one or a few individual loci. At the infrapopulation level, comparison of pre- and post-treatment populations showed that praziquantel did not substantially reduce

genomic diversity over a single round of treatment. We found minimal genomic differentiation between these populations suggesting post-treatment populations do not represent a distinct subpopulation. Across multiple studies, praziquantel treatment appears to have a variable impact on population genetic diversity with either no impact[23,24,67,78,79] or a reduction in diversity which recovered within months of treatment[20,22,80]. To our knowledge, no studies have shown long-term reductions in

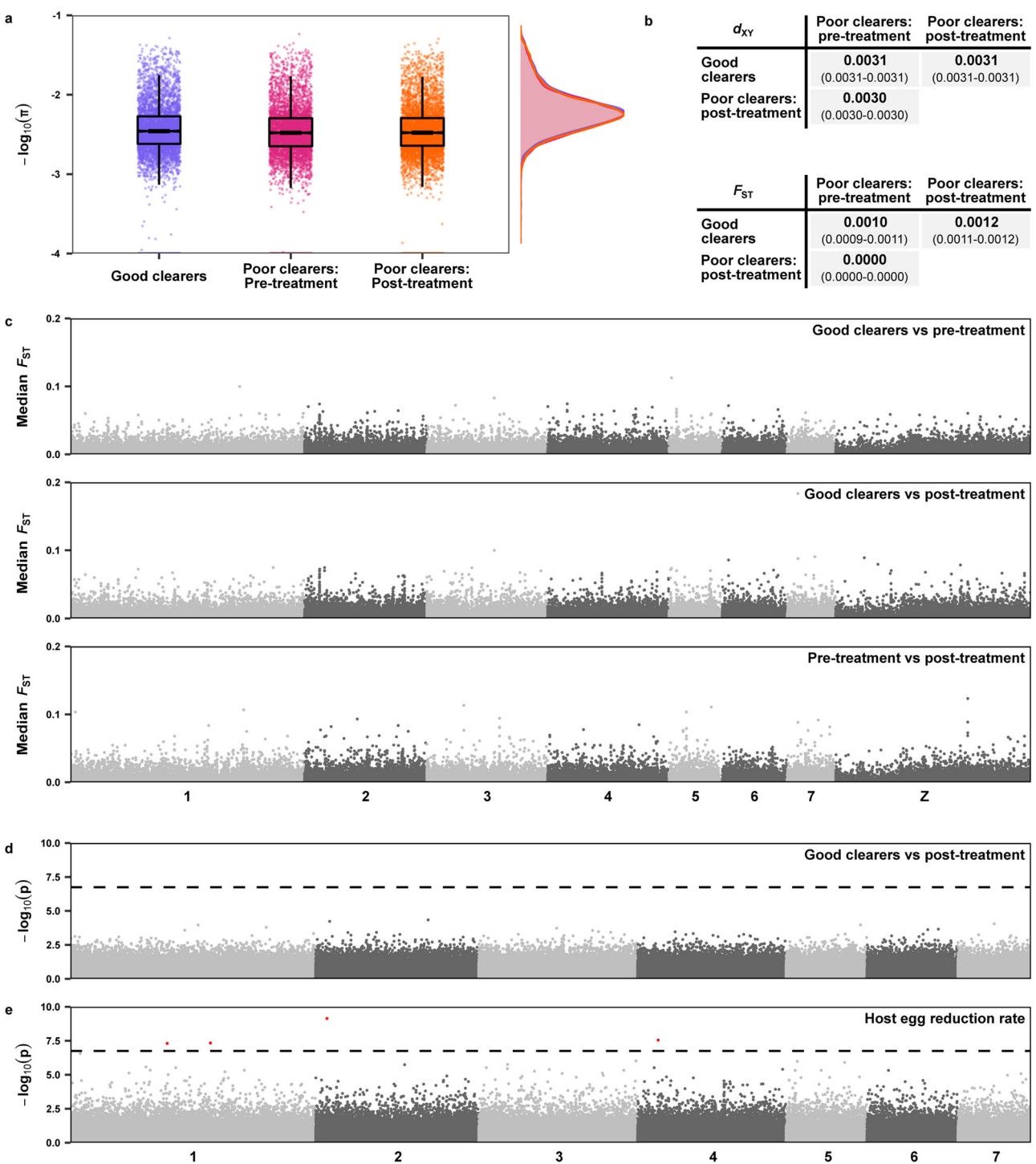

| $d_{XY}$ | Poor clearers: pre-treatment | Poor clearers: post-treatment |
| --- | --- | --- |
| Good clearers | **0.0031** (0.0031-0.0031) | **0.0031** (0.0031-0.0031) |
| Poor clearers: post-treatment | **0.0030** (0.0030-0.0030) | |

| $F_{ST}$ | Poor clearers: pre-treatment | Poor clearers: post-treatment |
| --- | --- | --- |
| Good clearers | **0.0010** (0.0009-0.0011) | **0.0012** (0.0011-0.0012) |
| Poor clearers: post-treatment | **0.0000** (0.0000-0.0000) | |

population genetic diversity in high prevalence settings. Van den Broeck et al.[81] recently modelled the potential impact of repeated praziquantel administration on schistosome populations. Their simulations suggested that repeated treatment could result in a measurable decline in genetic diversity at 95% treatment coverage. However, lasting declines were only predicted at 100% treatment coverage sustained over a minimum of six treatment rounds, a level of coverage that has not been attained in any endemic region.

Given the reduced praziquantel efficacy observed by Crellen et al.[42], we scanned the genomes for evidence of recent selection within Mayuge district, which could indicate adaptation to long-term MDA pressure. Evidence of positive selection was found in 25 regions of the *S. mansoni* genome, encompassing several genes from gene families suggested as potential mediators of praziquantel action or resistance. This included a gene encoding an ABC transporter that could represent a potential means of praziquantel efflux[82–85], and multiple calcium ion transporters that could regulate calcium influx caused by praziquantel[54–59]. However, many of these candidate genes were not found directly under the peak of these signals of positive selection and most functionally impactful variants were found at low frequencies

**Fig. 5 The impact of a single round of praziquantel treatment on Mayuge district _Schistosoma mansoni_ populations.** Populations were stratified into three groups based on clearance phenotype and time of sampling. Samples from children with no miracidial hatching after treatment were classified as 'good clearers' ($n = 81$, purple), samples from children with post-treatment miracidial hatching were grouped as either 'pre-treatment' ($n = 67$, pink) or 'post-treatment' ($n = 57$, orange) if they were sampled 25–27 days post-treatment. **a** Nucleotide diversity ($\pi$) estimates calculated in 5 kb, non-overlapping windows across each autosome for each population. For all boxplots, the central line indicates the median, the top and bottom edges of the box indicate the 25th and 75th percentiles, respectively. The maximum whisker lengths are specified as 1.5 times the interquartile range. **b** Pairwise comparisons of the fixation index ($F_{ST}$) and absolute divergence ($d_{XY}$) between populations, calculated in 5 kb, non-overlapping windows across all autosomes for each population. Numbers in bold represent the median value of all windows, numbers in parentheses represent the 95% bootstrap confidence intervals around the median. **c** Pairwise $F_{ST}$ estimates between each population, plotted as median values in 2 kb windows along each autosome (shown in alternating shades of grey). **d** Manhattan plot of significance values from logistic regression genome-wide association test, comparing populations sampled from good clearers to miracidia only from post-treatment collections. The test was performed using the 281,350 autosomal variants found not to be in strong linkage disequilibrium in these populations. **e** Manhattan plot of significance values from linear regression genome-wide association test using the quantitative egg reduction rate (ERR) estimates across all 181 Mayuge samples using the 281,954 autosomal variants not found to be in strong linkage disequilibrium. The first four principal components of the PCA (Fig. 4) were used as covariates. The dashed lines represent the Bonferroni corrected significance threshold, points higher than this threshold are shown in red.

within the Mayuge populations. Notably, we also found no evidence of selection acting on the $Ca^{2+}$-permeable transient receptor potential (TRP) channel (Smp_246790) recently identified as a likely target of praziquantel activity[60–62].

The simplest model of how selection leads to anthelmintic resistance would be for an initially rare resistance mutation at a single locus to be swept to fixation in a population (i.e. a hard selective sweep). Such events leave a strong signature in the genome around the locus, which can be detected by a number of population genetic methods, such as $F_{ST}$ and the haplotype tests that we employed in this study. Other scenarios, such as the presence of a resistance allele across a range of genetic backgrounds, which subsequently rise to higher frequencies (i.e. a soft sweep)[86], or polygenic traits are more difficult to detect and may have been missed by our study. This type of soft sweep is more likely to occur in a large and diverse population such as _S. mansoni_ around Lake Victoria and has been observed under similar scenarios in other helminth species[87–89]. If variants for praziquantel resistance are present across a range of genetic backgrounds, this could explain why genetic variation has been maintained over 10 years of MDA. Similarly, if praziquantel resistance is a polygenic trait with many mutations driving resistance, either additively or synergistically, the methods used in this study may be unable to detect such genes.

As MDA is not synchronised between the adjoining countries around Lake Victoria, these communities can miss praziquantel treatment rounds[90,91] and likely act as refugia of diverse parasites supporting the panmictic population structure. The high rates of gene flow we observed around Lake Victoria coupled with this diversity could also be preventing selection for or fixation of resistance-conferring variants. Even over the course of 8–9 years of MDA, post-treatment populations are small relative to the refugia populations in snails and untreated humans (adults, lightly infected individuals and asymptomatic carriers), and the high rates of gene flow between populations allow for rapid recovery of genetic diversity following treatment[48,92]. The longevity of adult schistosomes is likely to increase the effectiveness of these reservoirs of genetic diversity; even small numbers of sensitive adults surviving in refugia will produce eggs for many years. Management strategies for veterinary helminth infections often aim to create refugia populations to reduce the selection pressure for resistance[48,93,94] and it appears that the limited community coverage of MDA programmes may have had a similar effect. It is still possible that in the short-term, the long lifespan of adult schistosomes might enable praziquantel-resistant parasites to survive the entire duration of a school child's MDA, allowing them to act as a reservoir of resistance alleles[95,96]. However, over longer timescales, at present levels of MDA coverage, any

resistance-conferring variants are unlikely to be stably maintained in these populations, especially if they exhibit the reduced fitness found in some praziquantel-resistant _S. mansoni_ strains in the laboratory[97,98].

Praziquantel use also only represents a single selective pressure on Mayuge _S. mansoni_ populations. For example, inland and lakeshore _Biomphalaria_ are distinct populations and multiple _Biomphalaria_ species in Uganda can act as intermediate hosts for _S. mansoni_ cercariae (_B. sudanica_, _B. pfeifferi_, _B. choanomphala_)[99–101]. This structure has been found to influence infection prevalence in humans and could represent a differential selective pressure between districts[102]. Furthermore, while we found no indication of population structure by age or sex we do not know to what extent other host factors may be further stratifying our sample populations. Given the importance of identifying any evidence of reduced praziquantel susceptibility, we have focused our analyses on this but it is highly likely that _S. mansoni_ populations are responding and adapting to a diverse range of selective pressures.

Our findings have broad implications for future control efforts for schistosomiasis. As expected, our results show that MDA targeting school-age children has had a limited impact on the parasite population size or diversity at the community level. Our finding that the population of _S. mansoni_ in Mayuge remains essentially panmictic is in contrast to _Schistosoma japonicum_ populations in China, where one sign that elimination efforts have drastically reduced parasite circulation is the high level of population structure and relatedness between parasites from different hosts[27,103]. Together, these results suggest we are a long way from dramatically reducing schistosome transmission in this area of high infection prevalence and intensity. Mayuge district had received 8 or 9 rounds of treatment at the time our samples were collected, so this supports the growing consensus among the schistosomiasis control community that targeted MDA alone will be insufficient to interrupt schistosomiasis transmission[10]. We do not identify a single, strong signature of selection for praziquantel resistance, despite some evidence of reduced efficacy for the drug in this district. It could be that low efficacy is not a sign of resistance, but due to host or other factors[47,104–106], but it is also possible that multiple resistance-conferring alleles are present in the population—as appears to be the case for oxamniquine resistance[25]. If this is the case, we would expect resistance to emerge rapidly as the planned expansion and intensification of MDA coverage decreases the size of refugia, and therefore reduces gene flow that would dilute resistance-conferring alleles.

To conclude, we found that schistosome populations remain diverse and unstructured after nearly a decade of annual MDA and, at present, there appears to be no indication of a praziquantel-resistant subpopulation driving reduced clearance parasite

phenotypes. We did, however, identify a number of genomic regions with strong signals of selection that could indicate gradual adaptation to long-term praziquantel exposure or other selection pressures. Given the limited understanding of how natural populations of *S. mansoni* adapt to and are shaped by long-term MDA programmes we recommend further genomic surveillance across endemic regions. This surveillance would provide insights into the efficacy of current control strategies and allow the impact of selection on the parasite genome to be monitored providing advance warning of the emergence of anthelmintic resistance. Our study represents the largest WGS analysis of schistosome populations to date, and serves as a template for future work.

## Methods

**Sample collection and ethical approval**. Sampling was performed in Uganda, which was the first country in sub-Saharan Africa to start MDA against schistosomiasis[42,107]. This study was undertaken as part of monitoring and evaluation research activities conducted by the Schistosomiasis Control Initiative, Imperial College London and the Vector Control Division of the Ministry of Health, Uganda inherent with ongoing national disease control programme activities. All methods were approved by the Uganda National Council for Science and Technology (Memorandum of Understanding: sections 1.4, 1.5, 1.6) and the Imperial College Research Ethics Committee (EC NO: 03.36. R&D No: 03/SB/033E). This included the collection and publication of the indirect identifiers in Supplementary Data 9. Within Uganda, districts, sub-counties and schools were contacted about the nature of the fieldwork by the Head of the Vector Control Division, Ministry of Health Uganda. The head-teacher of the school was informed fully about the study and requested to provide informed consent, allowing the field-teams to collect samples from children within the school. Parents of children at the school were informed of the study through school meetings and were requested to provide informed consent for their children to participate within the study. Prior to consent, they were provided with detailed information as to why the study is taking place and any questions were answered by the authors and technical staff that were providing the information for the meeting. In addition to this, any child included who had reached age 10, was also asked to sign and give informed consent after receiving full information of the study. From those children from which their parents have provided informed consent, random selection was undertaken by the field-teams. Participation was voluntary, and children could withdraw or be withdrawn from the study at any time. Access to treatment was not dependent on consenting to participate in the study. All infected children were provided with praziquantel at 40 mg kg$^{-1}$.

We selected miracidia collected from children from four schools: three schools in Mayuge district that borders Lake Victoria—Bugoto ($n = 12$ children), Bwondha ($n = 10$ children) and Musubi ($n = 9$ children); and one school, Kocoge ($n = 3$ children), from Tororo district close to the Kenyan border (Fig. 1). These two districts, Mayuge and Tororo, were chosen as they had undergone long- and short-term MDA pressure, respectively (Table 1). For all sites, ~6 months had elapsed between the prior round of MDA and the pre-treatment sample collection for this study. Miracidia were isolated from infected children 1–3 days prior to treatment with praziquantel (40 mg/kg) for schistosomiasis, and albendazole (400 mg) for soil-transmitted helminths, and again in cases of incomplete clearance 25–27 days following treatment. The follow-up time frame was short to identify only treatment failures and reduce the probability of observing eggs from prepatent infections[108]. Stool samples were washed and filtered to retain parasite eggs with a Pitchford–Visser funnel. The filtrate was stored overnight in a petri-dish and hatched the next morning in sunlight. Miracidia were transferred in 3 µl of water into two sequential dishes of nuclease-free water to dilute bacterial contaminants and subsequently spotted, individually, onto Whatman FTA® cards.

Egg reduction rate (ERR) calculations for individuals and groups were estimated based on three days of duplicate pre- and post-treatment Kato-Katz slides using a mixed-effects regression model (https://github.com/jacotton/ERR_mixed_model)[42]. These scripts fit a Poisson generalized linear model to previously published pre- and post-sampling egg count data[109] with potential confounders included as main effects and school-level nested random effects using MCMCglmm (v.2.32)[110].

Parasite material was purposely selected following sample collection to represent a broad range of clearance phenotypes at the host level (Supplementary Data 9). Individuals were defined as having a 'good clearance' phenotype if no miracidia could be isolated from stool samples after treatment, or a 'poor clearance' phenotype if miracidia could be isolated following treatment. For poor-clearing individuals, it was therefore possible to obtain miracidia both before and after anthelmintic treatment. The posterior mean ERR for good clearing individuals ranged from 99.4 to 99.9% (median = 99.92%, 95% credible interval for median 99.85–99.98%) and for poor-clearing individuals from 24.2 to 99.4% (median = 90.93%, 95% credible interval for median 86.16–95.22%), calculated using the same model as in Crellen et al.[42] but including only those individuals for which miracidia were sequenced.

From 34 infected children, we aimed to sequence five randomly selected miracidia per child before treatment and, in the case of poor-clearing individuals, five randomly selected miracidia after treatment. In practice, there was not always sufficient parasite material, particularly from post-treatment egg collections, and so the number of miracidia at each timepoint varied from 1 to 9 (Fig. 1, Supplementary Data 9). In total we selected 222 miracidia, of which 204 were from 31 children in schools in Mayuge district (Bugoto $n = 86$, Bwondha, $n = 67$, Musubi $n = 51$) and 18 were from 3 children Kocoge school, in Tororo district. Overall 165 miracidia were sampled before treatment with praziquantel and 57 miracidia were sampled after treatment.

**DNA sequencing**. All sequenced samples were archived in the Schistosome Collection at the Natural History Museum (SCAN) prior to extraction[111]. Each miracidium was isolated from Whatman FTA® cards using a Harris 2 mm micro-punch and placed into 1.5 ml microcentrifuge tubes. DNA was extracted using a Qiagen DNeasy Blood and Tissue Extraction Kit following the manufacturer's instructions with the following alterations. 45 µl ATL buffer and 5 µl proteinase K were added to each tube and tubes were then vortexed, followed by incubation (56 °C on a rocking platform for 1 h). We added 50 µl of buffer AL and vortexed, then added 50 µl of 100% ethanol and the tubes were vortexed again. We removed 150 µl of liquid and pipetted it onto a spin column inside a collection tube. After centrifugation (6000 × g, 1 min), the collection tube was replaced and 500 µl of buffer AW1 was added to each spin column. After centrifugation (6000 × g, 1 min), the collection tube was replaced and 500 µl buffer AW2 was added to each spin column. After centrifugation (20,000 × g, 3 min), the collection tube was discarded and replaced with a 1.5 ml microcentrifuge tube. Fifty microliters of buffer AE (37 °C) was added to the spin column membrane and centrifuged (6000 × g, 1 min), an additional 50 µl of buffer AE (37 °C) was then added to the spin column membrane and centrifuged (6000 × g, 1 min).

DNA eluate was concentrated down to 10–50 µl using Savant SpeedVac (30 °C). Ten microliters of 2x denaturation buffer (GenomiPhi V3 amplification kit) was added and the tubes were heated (95 °C, 3 min) and cooled to 4 °C on ice. Denatured DNA was then added to the GenomiPhi freeze-dried pellet and incubated (30 °C, 90 min), before being cooled to 4 °C on ice. Amplified DNA was purified with magnetic AMPure XP beads and eluted in 50 µl nuclease-free water. Library preparation was performed by Wellcome Sanger Institute core facilities using NEBNext Ultra kit (New England Biolabs, E7370S) and KAPA HiFi HS ReadyMix (Kapa Biosystems, cat. no. KK2603) and either 8 or 18 PCR cycles were performed depending on library concentration Libraries were multiplexed and 150 bp paired-end reads were generated with the Illumina HiSeq X10 platform. Raw sequencing data generated during this study are available in the European Nucleotide Archive (ENA) repository under study accession number ERP113930 (Supplementary Data 2).

**SNP discovery and annotation**. The *S. mansoni* reference genome (v.7) was downloaded from WormBase Parasite[112,113] and scaffolds labelled as haplotype variants were removed using seqtk (https://github.com/lh3/seqtk; Supplementary Data 11). All sample reads were screened with Kraken (v.0.10.6) against a database of five standard datasets (Human, Mouse, Bacteria, Virus and Plasmid; Supplementary Data 12)[114]. Sequence reads were aligned using BWA mem (v.0.7.17) and duplicates were marked using PicardTools MarkDuplicates (as part of GATK v.4.1.0.0)[115,116]. Variant calling was performed per-sample using GATK HaplotypeCaller (v.4.1.0.0) in gVCF mode. Samples were then consolidated using GATK CombineGVCFs before joint-call cohort genotyping using GATK GenotypeGVCFs. GATK SelectVariants was used to separate single-nucleotide polymorphisms (SNPs) from indels and mixed sites before filtering using GATK VariantFiltration. SNPs were retained if they met the following criteria: QD ≥ 2.0, FS ≤ 60.0, MQ ≥ 40.0, MQRankSum ≥ −12.5, ReadPosRankSum ≥ −8.0, SOR ≤ 3.0 (Supplementary Fig. 2). Using VCFtools (v.0.1.15) we excluded samples with a high rate of SNP missingness (>55% of sites with a missing genotype), removed sites where >10% of samples had a missing genotype call, and finally removed all sites with a minor allele frequency (MAF) <1% (Supplementary Fig. 2, Supplementary Data 13)[117]. Variants found on the mitochondrial genome, unplaced contigs and W chromosome scaffolds were also excluded using VCFtools (Supplementary Data 11), leaving only variants found on the 8 chromosomal sequences (1–7 & Z) representing 98.56% of the assembly. Variant sites containing indels or mixed sites (variant sites that had both SNPs and indels called) were filtered independently and variants were retained if they met the following criteria: QD ≥ 2.0, FS ≤ 200.0, ReadPosRankSum ≥ −20.0, SOR ≤ 10.0. Samples excluded during SNP filtration were also excluded here (Supplementary Data 13). Functional annotation of SNPs and indels was performed using snpEff (v.4.3t) with gene annotations for *S. mansoni* (v.7) downloaded from WormBase Parasite.

**Depth of coverage**. We calculated the depth of read coverage in 2 and 25 kb windows along each chromosome using bedtools (v.2.30.0) coverage[118]. To differentiate male and female miracidia, we calculated median read coverage of the 2 kb windows across both pseudoautosomal regions of the Z chromosome: PAR1 (coordinates 1–10,739,103 bp) and PAR2 (coordinates 44,192,532–88,385,488 bp) and the non-recombining Z-specific region of the Z chromosome: ZSR (coordinates 10,739,104–44,192,531 bp). Samples

with a >0.75 ZSR/PAR ratio were designated as males and samples with <0.75 ZSR/PAR ratio were designated females.

**Sample relatedness**. We used KING (v.2.1.5) to calculate kinship coefficients and inferred identical-by-descent (IBD) segments between all pairwise combinations of samples[119]. Pairwise kinship coefficient ($\varphi$) scores of >0.354, 0.177–0.354, 0.0884–0.177 and 0.0442–0.0884 were used to classify duplicate/MZ twin, 1st-degree, 2nd-degree and 3rd-degree relationships, respectively. Relationships with PropIBD <0.0001 were excluded.

**Recombination**. The decay of linkage disequilibrium with genomic distance was determined for each population separately using VCFtools. The squared correlation coefficient was calculated between all sites within 50 kb and filtered to remove variants with minor allele frequency <5%. When populations had unequal numbers the larger population was randomly subsampled into multiple non-overlapping subpopulations equal to, or less than, the smallest population. Median values were calculated for each distance and plotted using the geom_smooth function of ggplot2 for each population, combining subsampled populations[120].

**Population genomic structure**. To analyse population structure, we used PLINK (v.2.0) to first remove variants found to be in strong linkage disequilibrium[121]. Variants were discarded according to the observed correlation between pairs of variant sites. The genome was scanned in sliding windows of 50 variants, in steps of 10 variants, and variants within windows with squared correlation coefficients ≥0.15 were removed (Supplementary Fig. 19). Variants found on the Z chromosome were also excluded. Principal component analysis (PCA) was performed using the remaining 1,344,965 variants using PLINK. Admixture analysis was performed using ADMIXTURE[122] with $K$ values (number of hypothetical ancestral populations) ranging from 1 to 20, 10-fold cross-validation, standard error estimation with 250 bootstraps and repeated the process 10 times with different random seeds. The lowest cross-validation error (CV) value was found for $K = 1$ increasing for each value of $K$ (Supplementary Fig. 5).

The 1,344,965 autosomal variants were used to construct a neighbour-joining tree. An identity-by-state distance matrix with distances expressed as genomic proportions was generated by PLINK. The ape (v.5.2) bionj algorithm was run on the resulting distance matrix[123]. The tree was midpoint rooted using phangorn (v.2.4.0) midpoint function and visualised using ggtree (v.1.10.5)[124,125].

PIXY (v.0.95.01) was used to calculate the nucleotide diversity ($\pi$), fixation index ($F_{ST}$) and absolute divergence ($d_{XY}$), in 5 kb sliding, non-overlapping windows across each autosome for each school and district population. Negative $F_{ST}$ values were corrected to 0 prior calculation of genome-wide median values for each population. Confidence intervals for these medians were calculated as the quantiles of the distribution of medians of 1000 bootstrap samples of genomic windows for each population.

**The impact of praziquantel treatment on population genomic diversity**. PIXY was used to calculate the nucleotide diversity ($\pi$), fixation index ($F_{ST}$) and absolute divergence ($d_{XY}$), in 5 kb, non-overlapping windows across each autosome for each treatment sampling stage and infrapopulation. For comparisons of infrapopulations, all samples from two children were excluded from $\pi$ analyses as well as $F_{ST}$ and $d_{XY}$ comparisons due to consistently low coverage across all infrapopulation samples (Supplementary Data 2). VCFtools was used to calculate the fixation index ($F_{ST}$) in 2 kb windows for each treatment sampling stage. We tested for significant differences between male and female infrapopulations using a two-tailed $t$-test comparing median autosomal nucleotide diversity for each infrapopulation. Pearson correlation coefficients and $p$-values analysing the relationship between age or infrapopulation size were calculated using the ggpubr (stat_cor) function using default parameters[127].

**Detection of genome-wide signatures of selection**. Statistical phasing was performed using Beagle (v.5.0) and a recombination map based on per-chromosome recombination rate estimates from Criscione et al.[128,129]. The effective population size was set at 65,000 based on estimates by Crellen et al.[17]. 4,428,787 phased bi-allelic variants were then used to calculate the integrated haplotype scores (iHS) and cross-population extended haplotype homozygosity (XP-EHH) using Selscan (v.1.2.0a) for each chromosome separately[130]. Normalisation was performed for all Selscan outputs across all chromosomes using norm (v.1.2.0a) distributed with Selscan[130]. Median iHS and XP-EHH scores were calculated for 2 kb non-overlapping windows along each chromosome. VCFtools was used to calculate fixation index ($F_{ST}$) and $\pi$ in 2 kb sliding, non-overlapping windows across each chromosome. The ratio in nucleotide diversity was calculated as the median $\pi$ value within each 2 kb window between Mayuge and Tororo district populations. For all four statistics windows with fewer than 20 variants per 2 kb window were removed ($n = 3207$). Consistent signals across multiple statistics were taken as strong evidence of selection[131]. The highest 0.25% of Mayuge |iHS|, XP-EHH and $F_{ST}$ windows were considered candidate regions of selection. Any windows within 300 kb of another were joined into single continuous regions of

selection, these were then manually inspected, extending regions where peak structure was not fully encompassed. Any windows found within the Z-specific region (located between 10.74 and 44.19 Mb) on the Z chromosome were not included as candidate regions of selection.

As additional supporting evidence we calculated Tajima's D statistic in 2 kb non-overlapping windows using VCFtools. As above the same windows with fewer than 20 variants per 2 kb were removed. In addition, per variant site $F_{ST}$ and $\pi$ values were calculated using VCFtools (Supplementary Data 14).

**Association testing**. Association tests were performed following removal of variants in close linkage as described above. In addition, variants with a minor allele frequency <5% were then removed. Two different association tests were performed. First, a logistic regression genome-wide association study was performed between samples from children with 'good clearance' phenotypes and post-treatment samples from children with poor clearance phenotypes (using 281,350 variants). Second, a linear regression genome-wide association study was conducted in PLINK (v.1.9) with the ERR estimates for all 198 samples, using the mean of the posterior ERR estimates from Crellen et al.[17] as a quantitative trait (using 281,954 variants). For both tests, the first four principal components of the PCA were used as covariates, and a Bonferroni corrected threshold was used to determine significance.

**Inference of population history**. We used SMC++ (v1.15.2) to estimate the size history of each population[132]. SMC++ was run on each autosome using a per-generation mutation rate of 8.1e−9 and a generation time of 71 days[17]. All samples from one population (for each school or, in the case of outgroups, each country) were included. Subsampled replicates were produced by randomly subsampling 7.5 Mb from each chromosome 25 times for each population. One-dimensional site frequency spectrum was calculated using easySFS (https://github.com/isaacovercast/easySFS), we projected down to 15 samples per population and plotted estimates produced by δaδi (v.2.7.0)[133]. This was repeated for a total of five replicates. For analyses of recent demographic history, the Tajima's D statistic was calculated in 5 kb non-overlapping windows using VCFtools.

**Software**. For all software, unless otherwise stated, default parameters were used. All figures were produced with R (3.5.1), ggplot2 (v.3.1.0) and cowplot[120,134,135]. A map of sampled regions was generated with QGIS (v.3.2.2) using raster and vector map data from Natural Earth (found at naturalearthdata.com). GNU parallel[136], GNU datamash (v1.4, gnu.org/software/datamash) and BCF tools[137] were used at multiple stages to facilitate data processing and analyses.

**Reporting summary**. Further information on research design is available in the Nature Research Reporting Summary linked to this article.

## Data availability

The sequencing data generated in this study have been deposited in the European Nucleotide Archive (ENA) repository under accession code PRJEB31375. Individual sample accessions are listed in Supplementary Data 10. Source data are provided as a Source Data folder and are also available at [https://doi.org/10.5281/zenodo.4940588][138]. Genome and annotation files are available through WormBase Parasite [https://parasite.wormbase.org/Schistosoma_mansoni_prjea36577/Info/Index/]. Egg count data used to produce the egg reduction rate estimates are available [10.13140/RG.2.2.12687.84640]. To reduce the number of indirect identifiers available, children's ages removed, access can be obtained upon reasonable request to the corresponding authors. Source data are provided with this paper.

## Code availability

The code used for data analysis is available at [https://doi.org/10.5281/zenodo.4975908][139].

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

## Acknowledgements

Funding was via core funding of the Wellcome Sanger Institute (Wellcome grants 098051 and 206194), by support of T.C. via an MRC DTP Studentship (under the supervision of J.P.W. and J.A.C.), and fieldwork was supported within the remit of ongoing Schistosomiasis Control Initiative (SCI) monitoring and evaluation activities. F.A. and the Schistosome Collection at the Natural History Museum (SCAN) were supported by Wellcome grant 1045958/Z/13/Z, We thank the technicians, nurses, and drivers from the Vector Control Division, Ugandan Ministry of Health, Moses Arinaitwe, Aida Wamboko, Annet Enzaru, Andrina Nankasi, Aaron Atuhaire and Fiddi Rugigana, whose hard work in data collection was instrumental to the study; the teachers from the schools we visited for their assistance and cooperation along with officials in the Health and Education Departments in districts of Mayuge and Tororo, Uganda, for their cooperation with the study; and Alan Fenwick and Wendy Harrison of the Schistosomiasis Control Initiative for their assistance in organizing and funding the fieldwork. We would also like to thank Steve Doyle and Sarah Buddenborg for their helpful feedback on earlier versions of this manuscript. Finally, we thank the children recruited into this study and their parents and teachers for their cooperation.

## Author contributions

T.C., M.B., J.P.W. and J.A.C. conceived of the project, for which M.B. and J.P.W. secured funding. T.C., N.B.K., E.M.T., P.H.L.L. and J.P.W. planned and coordinated the fieldwork, which was carried out by T.C., P.H.L.L. and M.A. N.H. planned and coordinated the molecular biology and sequencing. D.J.B. and T.C. analysed data with input from A.T. and J.N., T.C. and F.A. extracted and amplified, DNA. D.J.B., T.C., M.B., J.P.W. and J.A.C. wrote the paper with input and approval from all authors.

## Competing interests

The authors declare no competing interests.
