## [Peer Review File · Nature Communications]

Reviewers' Comments:

Reviewer #1:

Remarks to the Author:

General comments

The authors are to be commended on generating a large and significant set of genome sequences from an important neglected pathogen in a well defined epidemiological setting that is already the subject of other non-genetic investigations. In particular, these data and the analyses of them contained in this manuscript extend the appreciation of the very large population sizes and correspondingly very high degree of genetic diversity of these metazoan parasites, and the impact that large N_e and high genetic diversity have on our ability to analyse these parasites genetically. In particular, the confirmation of earlier more limited studies on genetic diversity and lack of fine scale population structure in *S. mansoni* is important, as is their observation that several years of MDA have made no detectable impression on this genetic diversity.

However, I found the authors' focus on what appears to be a simplistic model for resistance selection both frustrating and surprising. Frustrating because although the authors acknowledge that there is no support for resistance selection in these populations, the discussion of the possible reasons for this failure are largely technical (sample size, low statistical power due to a combination of small sample size and high genetic diversity, large genome size etc) and does not propose alternative genetic hypotheses are supported by the data i.e. that sensitivity to PZQ is a continuously variable quantitative trait where many alleles contribute additively to the phenotype. This is a contrast between a hypothesis () of hard selection for a single gene with major effect (the candidates discussed) versus soft selection of a quantitative trait from standing genetic variation in a very large, very diverse population. Surprising because one of the authors (Cotton, in Doyle and Cotton, 2019, Trends in Parasitology, doi.org/10.1016/j.pt.2019.01.004) recently published a review making just this point i.e. drug susceptibility as a quantitative trait with soft sweeps that are difficult to detect).

At the very least, the authors should be more explicit in enunciating their resistance hypothesis in the theoretical and methodological framework described by Doyle & Cotton.

The authors also, in my view, miss an opportunity then to extend our understanding of the population genetics of drug response in helminth parasites more broadly and make little or no reference to work in other helminths (filaria, veterinary helminths) for which comparable and in some cases, more extensive, data exist. The authors make passing reference to refugia, and it is here in particular there a missed opportunity. The MDA program for schistosomiasis is, if I understand correctly, targeted at school age children but not at teenagers and adults who are also infected. Thus there are at least two refugia populations: the untreated human hosts in the same communities and the parasites that are present in the snails. Schistosomes are also notable for the longevity of the adult worms, so that a small sub-population of genetically diverse resistant parasites might survive for the entire duration of a schoolchild's MDA, but with a presumably reservoir of refugia parasites, there will be relatively little actual selection and little or no turnover of the resistant population in the targeted children. This work would be more likely to have some impact if the authors explored the epidemiological and population genetic implications of their data in this direction rather than maintain a largely negative focus on the failure to detect selection that is based on a narrow hypothesis of hard selection.

There is also a more general risk in this focus. As the authors point out, the question of PZQ resistance is controversial despite evidence of variation in efficacy. As presented, this study may give support to a view that resistance is unlikely to evolve and that further investigation is not warranted. I realise that this is very clearly not the authors' intention, and propose that such an interpretation may be less likely if the authors were to focus more on the contribution their data may make to better understanding the genetic epidemiology and population dynamics of *S. mansoni*. In particular, the data support a view that MDA programs such as those analysed in this study, are unlikely to have long term sustainable impacts on the parasite population. The failure to detect a reduction in population genetic diversity after 10 years of MDA in this study, even allowing for the relatively small sample size, leaves no alternative interpretation other than a failure of the MDA to significantly reduce the parasite population size. This implies that

recrudescence will occur if MDA is stopped. There is little doubt that there are benefits of MDA to those individuals that receive treatment, but the evidence presented here suggests that there is unlikely to be a sustained impact on schistosome transmission and epidemiology with this form of targeted MDA.

I do not think this manuscript is of sufficient significance in its current form for publication in Nature Communications. I think this is a pity given that the genome sequence data that are the basis of this manuscript are significant. Refocusing the paper to give less emphasis to resistance (particularly the narrow hard selection hypothesis that is implicit in the manuscript) and more emphasis to the genetic epidemiological implications of the data (and the possibility for soft selection of quantitative variation in PZQ response) may redeem the situation.

Specific comments:

I have not collated specific comments for the authors because I do not think the manuscript is publishable with its current focus.

(Reviewed by Warwick Grant)

Reviewer #2:

Remarks to the Author:

In this paper, the authors analyze 198 whole-genome sequences of *Schistosoma mansoni*, the parasite that causes schistosomiasis. Samples had been collected from children from several schools in two Ugandan districts with varying histories of drug (praziquantel) treatment. The authors find mild genetic differentiation between the districts, but little differentiation among hosts within districts, and similar levels of genetic diversity across all samples. They identify several putative signals of positive selection and a few significant associations between genotype and egg reduction rate, but no particularly compelling candidate genes.

This is a thorough analysis of a large dataset which will be a standard for other schistosome researchers going forward. The various bioinformatic tests are appropriate and performed correctly. Essentially this paper represents a negative result, in that the authors were presumably hoping to find a clear signal of selection by praziquantel, pointing to candidate genes for resistance. However, there is no "smoking gun", and there is little evidence that praziquantel has played an important role in shaping the observed genetic signals. In a few places, including the title, the paper seems to be overly framed around praziquantel. The "impact" promised by the title remains elusive at best in the data. Still, throughout most of the paper, the authors are appropriately cautious in their interpretation of suggestive genomic regions. And regardless of any relation to praziquantel, this paper provides insight into overall evolutionary genetic patterns in this important parasite, with the final sentence of the Abstract summarizing the key value of these results.

There is no compelling reason to conclude that the observed signals of selection are due to praziquantel specifically. As a control, it would be interesting to see if there is more evidence for selection in Mayuge than Tororo, but the authors did not look for Tororo-specific selection (and there might be low power to do so for some tests given the relatively small sample size there). If there has been selection on standing variation in Mayuge, but much weaker selection in Tororo, one should expect to see reduced genetic diversity surrounding the targeted locus in Mayuge relative to Tororo. The authors calculate genetic diversity in windows across the genome (Fig 2C) but they never compare across districts for each window. If there is a genomic region with a substantially reduced Mayuge/Tororo ratio of π , especially if there is no comparable region with a reduced Tororo/Mayuge ratio, that could be a more convincing signal of praziquantel-driven selection. Certainly that would be a more expected outcome of selection than total reduction in π across the genome (which would only happen in an extreme case), so the authors should spend at least as much time looking for locus-specific reductions in π as they do looking for genome-wide reductions.

Statistics like F_{ST} , XP-EHH, and iHS can be applied to individual variants, not just chromosomal

regions. They are the most powerful when applied in this way. If the authors are going to highlight particular variants as interesting (e.g. lines 388-389, 400-403), it would be good to indicate whether those specific variants show signals of selection, not just the regions in which they are found. Relatedly, especially given the decay of LD (Supp Fig 2), 25 kb may be too wide of a window to capture the signal of these stats. There could be strong signals of selection that are being diluted by this wide-window approach. For example, if there are windows with median F_{ST} over 0.5, then possibly there are individual variants that are nearly or completely fixed (i.e. $F_{ST} > 0.95$). If any of these occur in genes, it would be interesting to know what they are.

Abstract: The phrase "encompassing genes previously linked to praziquantel action" is overstated. Some of the outlier genes are in broad gene families (i.e. ABC transporter or calcium exchange) that have been hypothesized as possibly involved in praziquantel resistance. That's not the same as saying these specific genes are linked to the drug.

Is there any correlation between signals of selection and read depth (shown in Supp Figure 4)? Specifically, repetitive regions can show unusually high depth (due to misaligned paralogs) and also causes genotyping artifacts that can be interpreted as selection signals.

Figure 4C & 4D: The authors are appropriately cautious about the outliers differentiated between "good clearers" and "post treatments" and dismiss them as probably not biologically meaningful. I am inclined to agree, but it wouldn't hurt to include a few of the top outliers in a supplementary table to show their specific locations. In particular, I found myself wondering about the outlier(s) at the start of chromosome 2, and whether they overlap with signals of selection in that vicinity.

Reviewer #3:

Remarks to the Author:

This is a well-written and technically high quality investigation which is worthy of consideration of Nature Comms, however, there are some fundamental weaknesses which make me less inclined to recommend publication without revision(s) and better discussion of bias within the inspected population of miracidia.

1. The sampling design is ad hoc and by convenience. Effects for example of hatchability of miracidia were ignored, I was disappointed not to see specific mention of how many male or female miracidia were collected given the genotyping of ZZ and ZW should be able to investigate this. Was there a bias towards male miracidia?

I am therefore concerned that although the authors have done their best, an in-depth examination from 34 children and 222 (198) miracidia does have weaknesses when attempts are made to broaden their findings. I find it somewhat confusing with statements about panmictic populations and then 'localized' selection effects largely thought to come about by drug induced selection or low cure. Is it not more interesting in that the diversity is so massive it likely swamps any attempts to dampen it from any human based intervention?

Could the authors provide more details about the choice of the children and why they were selected, and how they may or may not represent a broader child population with the schools. Are they sure for example the children were local to the sampled schools or were more itinerant? More convincing is needed.

2. The genetic diversity measures are fascinating and interesting in their own right as a 'microcosm' of what is going on within a child, however, there is complete oversight how these genotypes partition through time in the local snail population. It is a shame that at least a handful of local cercariae were not sampled which could really tie-down their ascertainment of panmixia for one would suspect (but as of yet not confirmed) that many of these genotypes would be locally present. It is therefore an open question which for journals other than Nature Comms could remain speculative but in this instance needs some due consideration and diligence.

3. The divergence between the local Lake Victoria *S. mansoni* population and the one more inland

as a comparator, also have some bearing on 2. For example, *Biomphalaria* inland is not the same as those populations in the lake. It is likely that a reason why there is divergence could be due to local snail-parasite selection to *B. pfeifferi* versus *B. choanomphala*. This adds another level of uncertainty to the mechanisms by which the genetic diversity of these parasites are shaped. Greater consideration of the biology of transmission is needed to balance out the emphasis on praziquantel selection pressure. It is a shame there is no further genetic sampling of *S. mansoni* populations inland. Is it not possible to look at other *S. mansoni* data to compare to bolster this as it is a weakness in my mind when there is an inference across a broader spatial level. Some reference to <https://parasitesandvectors.biomedcentral.com/articles/10.1186/s13071-014-0524-4> might help better position their speculations.

4. I commend the authors on their bioinformatics and associated redaction of their sequence information. This adds clear value to the 'baseline' diversity. By inference, with such diversity in a small place, what expectations are there for the *S. mansoni* Lake Victoria populations as a whole? It is possible to make some speculations on the effective population size given F_{st} measures have been made and it is possible to make some inferences on minimum estimate which would satisfy infinite population assumptions. I think it important to tone down the search for genetic resistance and concentrate more on the fascinating population genetics which is coming to light which is quite different to how we might have originally perceived it.

Responses to reviewers

Reviewer #1:

General comments

The authors are to be commended on generating a large and significant set of genome sequences from an important neglected pathogen in a well defined epidemiological setting that is already the subject of other non-genetic investigations. In particular, these data and the analyses of them contained in this manuscript extend the appreciation of the very large population sizes and correspondingly very high degree of genetic diversity of these metazoan parasites, and the impact that large N_e and high genetic diversity have on our ability to analyse these parasites genetically. In particular, the confirmation of earlier more limited studies on genetic diversity and lack of fine scale population structure in *S. mansoni* is important, as is their observation that several years of MDA have made no detectable impression on this genetic diversity.

However, I found the authors' focus on what appears to be a simplistic model for resistance selection both frustrating and surprising. Frustrating because although the authors acknowledge that there is no support for resistance selection in these populations, the discussion of the possible reasons for this failure are largely technical (sample size, low statistical power due to a combination of small sample size and high genetic diversity, large genome size etc) and does not propose alternative genetic hypotheses are supported by the data i.e. that sensitivity to PZQ is a continuously variable quantitative trait where many alleles contribute additively to the phenotype. This is a contrast between a hypothesis () of hard selection for a single gene with major effect (the candidates discussed) versus soft selection of a quantitative trait from standing genetic variation in a very large, very diverse population. Surprising because one of the authors (Cotton, in Doyle and Cotton, 2019, Trends in Parasitology, doi.org/10.1016/j.pt.2019.01.004) recently published a review making just this point i.e. drug susceptibility as a quantitative trait with soft sweeps that are difficult to detect).

At the very least, the authors should be more explicit in enunciating their resistance hypothesis in the theoretical and methodological framework described by Doyle & Cotton.

We agree and the text has been rephrased and refocused accordingly. Our original design was aimed to identify any genomic evidence to support the observed phenotypic reductions in drug efficacy in response to longer term praziquantel pressure we have previously reported from these same populations³. We appreciate that part of the reason this may not have been reflected here could be due to the genetic architecture of the resistance trait, as well as the technical limitations discussed. We have restructured and expanded our focus accordingly overall. Likewise, we have also brought in consideration of the theoretical and methodological frameworks recently proposed in Doyle and Cotton and we thank the referee for this advice.

The authors also, in my view, miss an opportunity then to extend our understanding of the population genetics of drug response in helminth parasites more broadly and make little or no reference to work in other helminths (filaria, veterinary helminths) for which comparable and in some cases, more extensive, data exist. The authors make passing reference to refugia, and it is here in particular there a missed opportunity. The MDA program for schistosomiasis is, if I understand correctly, targeted at school age children but not at

teenagers and adults who are also infected. Thus there are at least two refugia populations: the untreated human hosts in the same communities and the parasites that are present in the snails.

We have expanded our discussion of *refugia* and included more references to other helminths. (Though also to note, depending on baseline endemicities, MDA for schistosomes is either school-aged-children or community-wide treatment, but we appreciate this issue regarding the populations examined here).

Schistosomes are also notable for the longevity of the adult worms, so that a small sub-population of genetically diverse resistant parasites might survive for the entire duration of a schoolchild's MDA, but with a presumably reservoir of refugia parasites, there will be relatively little actual selection and little or no turnover of the resistant population in the targeted children. This work would be more likely to have some impact if the authors explored the epidemiological and population genetic implications of their data in this direction rather than maintain a largely negative focus on the failure to detect selection that is based on a narrow hypothesis of hard selection.

The discussion has been revised to include a more in-depth consideration of *refugia*, including the impact of parasite longevity.

There is also a more general risk in this focus. As the authors point out, the question of PZQ resistance is controversial despite evidence of variation in efficacy. As presented, this study may give support to a view that resistance is unlikely to evolve and that further investigation is not warranted. I realise that this is very clearly not the authors' intention, and propose that such an interpretation may be less likely if the authors were to focus more on the contribution their data may make to better understanding the genetic epidemiology and population dynamics of *S. mansoni*.

We thank this referee for flagging these omissions and, critically, the potential for misinterpretation. We have substantially revised our discussion. We have retained the suggestion that the *refugia* populations (and high rates of gene flow) could be preventing the retention or fixation of resistance alleles in *S. mansoni* populations, which has been suggested by other publications⁴⁻⁶. We have, however, clarified that this does not imply resistance will not emerge with expansions of MDA coverage particularly given the high level of natural variation and the possibility that praziquantel resistance is a quantitative trait.

In particular, the data support a view that MDA programs such as those analysed in this study, are unlikely to have long term sustainable impacts on the parasite population. The failure to detect a reduction in population genetic diversity after 10 years of MDA in this study, even allowing for the relatively small sample size, leaves no alternative interpretation other than a failure of the MDA to significantly reduce the parasite population size.

As above, we have expanded our discussion accordingly. We have also included additional analysis of population size. Although emergence from a bottleneck appears possible, the effect is not specific to the Mayuge population and it is therefore unlikely that MDA has had a sustained impact on these populations.

This implies that recrudescence will occur if MDA is stopped. There is little doubt that there are benefits of MDA to those individuals that receive treatment, but the evidence presented here suggests that there is unlikely to be a sustained impact on schistosome transmission and epidemiology with this form of targeted MDA.

As evidenced by the new WHO NTD 2021-2030 Roadmap ², the predominant viewpoint appears to be that targeted MDA alone (particularly if encompassing school-aged children only) will be insufficient to interrupt schistosomiasis transmission, or even elimination as a public health problem. We believe that our findings here lend further support for the WHO's revised strategies and guidelines - and this has been accommodated into our revised text

I do not think this manuscript is of sufficient significance in its current form for publication in Nature Communications. I think this is a pity given that the genome sequence data that are the basis of this manuscript are significant. Refocusing the paper to give less emphasis to resistance (particularly the narrow hard selection hypothesis that is implicit in the manuscript) and more emphasis to the genetic epidemiological implications of the data (and the possibility for soft selection of quantitative variation in PZQ response) may redeem the situation.

We trust you agree that the manuscript has now been suitably refocused and expanded to satisfy the points raised above and we thank you again for your valid points raised.

Reviewer #2:

In this paper, the authors analyze 198 whole-genome sequences of *Schistosoma mansoni*, the parasite that causes schistosomiasis. Samples had been collected from children from several schools in two Ugandan districts with varying histories of drug (praziquantel) treatment. The authors find mild genetic differentiation between the districts, but little differentiation among hosts within districts, and similar levels of genetic diversity across all samples. They identify several putative signals of positive selection and a few significant associations between genotype and egg reduction rate, but no particularly compelling candidate genes.

This is a thorough analysis of a large dataset which will be a standard for other schistosome researchers going forward. The various bioinformatic tests are appropriate and performed correctly. Essentially this paper represents a negative result, in that the authors were presumably hoping to find a clear signal of selection by praziquantel, pointing to candidate genes for resistance. However, there is no "smoking gun", and there is little evidence that praziquantel has played an important role in shaping the observed genetic signals. In a few places, including the title, the paper seems to be overly framed around praziquantel. The "impact" promised by the title remains elusive at best in the data. Still, throughout most of the paper, the authors are appropriately cautious in their interpretation of suggestive genomic regions. And regardless of any relation to praziquantel, this paper provides insight into overall evolutionary genetic patterns in this important parasite, with the final sentence of the Abstract summarizing the key value of these results.

Thank you and we agree that, in addition to any potential relationship, or not, with praziquantel, our study provides broader insight into the evolutionary genetics of this

important parasite. We have, as detailed above, refocused our text (including the title and incorporating additional data analyses) accordingly.

There is no compelling reason to conclude that the observed signals of selection are due to praziquantel specifically. As a control, it would be interesting to see if there is more evidence for selection in Mayuge than Tororo, but the authors did not look for Tororo-specific selection (and there might be low power to do so for some tests given the relatively small sample size there).

Agreed. In addition to our revised text, as above, we have now also included the $|iHS|$ values for the Tororo samples in the main text (Figure 4). We have also included Tajima's D comparisons between Mayuge and Tororo (Supplementary Figure 11).

If there has been selection on standing variation in Mayuge, but much weaker selection in Tororo, one should expect to see reduced genetic diversity surrounding the targeted locus in Mayuge relative to Tororo. The authors calculate genetic diversity in windows across the genome (Fig 2C) but they never compare across districts for each window. If there is a genomic region with a substantially reduced Mayuge/Tororo ratio of π , especially if there is no comparable region with a reduced Tororo/Mayuge ratio, that could be a more convincing signal of praziquantel-driven selection. Certainly that would be a more expected outcome of selection than total reduction in π across the genome (which would only happen in an extreme case), so the authors should spend at least as much time looking for locus-specific reductions in π as they do looking for genome-wide reductions.

Agreed and thank you for raising this - we have now included $\pi_{\text{Mayuge}}/\pi_{\text{Tororo}}$ as part of Figure 4 and Supplementary figures 13E-18E.

Statistics like F_{ST} , XP-EHH, and iHS can be applied to individual variants, not just chromosomal regions. They are the most powerful when applied in this way. If the authors are going to highlight particular variants as interesting (e.g. lines 388-389, 400-403), it would be good to indicate whether those specific variants show signals of selection, not just the regions in which they are found.

Agreed. We have included additional figures in our revised Supplementary Information (Supplementary figures 13-18) showing per-site values for each statistic over many of the candidate selected regions (including all regions discussed in the main text).

Relatedly, especially given the decay of LD (Supp Fig 2), 25 kb may be too wide of a window to capture the signal of these stats. There could be strong signals of selection that are being diluted by this wide-window approach.

Agreed. We have reduced the window size to 2 kb for all statistics. Whilst ideally, we would show each variant across all chromosomes, the large number of variants makes plotting this extremely impractical, and thus, as mentioned above, we have instead shown per-variant statistics for each candidate selected region that we discuss in the main text.

For example, if there are windows with median F_{ST} over 0.5, then possibly there are individual variants that are nearly or completely fixed (i.e. $F_{ST} > 0.95$). If any of these occur in genes, it would be interesting to know what they are.

We have now included an additional Supplementary table showing all F_{ST} scores for all variant sites where $F_{ST} \geq 0.9$ (Supplementary Table 14)

Abstract: The phrase “encompassing genes previously linked to praziquantel action” is overstated. Some of the outlier genes are in broad gene families (i.e. ABC transporter or calcium exchange) that have been hypothesized as possibly involved in praziquantel resistance. That’s not the same as saying these specific genes are linked to the drug.

Agreed and the text rephrased accordingly.

Is there any correlation between signals of selection and read depth (shown in Supp Figure 4)? Specifically, repetitive regions can show unusually high depth (due to misaligned paralogs) and also causes genotyping artifacts that can be interpreted as selection signals.

This is an excellent question, however, given the large number of samples and generally variable per-sample read depth it’s extremely difficult to correlate read depth and selection statistics (either as a population or per sample per statistic). That said, we have shown relative coverage (Mayuge/Tororo) in each candidate region of selection we discuss in the main text (Supplementary figures 13F-18F) to identify any regions of differential coverage which could then drive differential signatures of selection (although we also acknowledge that this is not an ideal metric).

Figure 4C & 4D: The authors are appropriately cautious about the outliers differentiated between “good clearers” and “post treatments” and dismiss them as probably not biologically meaningful. I am inclined to agree, but it wouldn’t hurt to include a few of the top outliers in a supplementary table to show their specific locations. In particular, I found myself wondering about the outlier(s) at the start of chromosome 2, and whether they overlap with signals of selection in that vicinity.

Agreed and these have now been included in an additional table within the Supplementary information with corresponding text (Supplementary Table 8). None of the outliers occurred within candidate selected regions (Supplementary Tables 4 and 5): the outlier on chromosome 2 occurred at position 2.523377 Mb the nearest selected regions was between 2.624001–2.756001 Mb.

Reviewer #3:

This is a well-written and technically high-quality investigation which is worthy of consideration of Nature Comms, however, there are some fundamental weaknesses which make me less inclined to recommend publication without revision(s) and better discussion of bias within the inspected population of miracidia.

1. The sampling design is ad hoc and by convenience. Effects for example of hatchability of miracidia were ignored, I was disappointed not to see specific mention of how many male or

female miracidia were collected given the genotyping of ZZ and ZW should be able to investigate this. Was there a bias towards male miracidia?

We have now included estimated counts of male and female miracidia (Supplementary figure 3 and Supplementary figure 9). Due to the relatively high similarity between Z and W gametologues and the highly repeat content of the W-specific (non-recombining) region of the W chromosome we identified males and females based on coverage over the PAR and Z-specific regions of the Z chromosome ⁷. We identified 103 miracidia samples as female and 95 as males (Supplementary Figure 9, Supplementary Table 3).

I am therefore concerned that although the authors have done their best, an in-depth examination from 34 children and 222 (198) miracidia does have weaknesses when attempts are made to broaden their findings. I find it somewhat confusing with statements about panmictic populations and then 'localized' selection effects largely thought to come about by drug induced selection or low cure. Is it not more interesting in that the diversity is so massive it likely swamps any attempts to dampen it from any human based intervention?

Agreed and our text has been rewritten accordingly. We believe, at least in the early stages of control programmes under such high transmission regions, that the substantial levels of genomic diversity and gene flow observed may be sufficient to dampen evidence of any impact of human-based intervention. However, we feel that this may then change once expanded schistosomiasis MDA programmes begin to near the elimination targets, and we have thus elaborated further upon this changing dynamic within our re-written discussion.

Could the authors provide more details about the choice of the children and why they were selected, and how they may or may not represent a broader child population with the schools. Are they sure for example the children were local to the sampled schools or were more itinerant? More convincing is needed.

As this genomic study followed on from the study of Crellen et al. ³, we referred readers to the original paper for fuller details. The children were registered and hence representative, from within the sampled schools and selected regions.

We sequenced parasites from children representing a broad range of clearance phenotypes (egg reduction rates) and where sufficient parasite samples were collected to enable five miracidia to be sequence pre-treatment and (where appropriate) five miracidia post-treatment. We also aimed to get good representation of parasite populations from each school, but in the case of Kocoge school (Tororo district), low numbers of sampled parasites (due to overall lower prevalence of infections) limited us.

We don't have a great deal of information about the migration of children sampled between schools, districts or countries. It is known that a high proportion of adults in these shoreline communities work in the fishing industry and this can result in the movement of families around Lake Victoria ⁸. However, information about the attendance, or migratory history of the children sampled is not available to us.

2. The genetic diversity measures are fascinating and interesting in their own right as a 'microcosm' of what is going on within a child, however, there is complete oversight how these genotypes partition through time in the local snail population. It is a shame that at least a handful of local cercariae were not sampled which could really tie-down their ascertainment of

panmixia for one would suspect (but as of yet not confirmed) that many of these genotypes would be locally present. It is therefore an open question which for journals other than Nature Comms could remain speculative but in this instance needs some due consideration and diligence.

We agree that to comprehensively elucidate the schistosome population structure in any endemic environment, the ideal would be to obtain material from across all life stages of the parasite. Free-living larval stages (miracidia, as used here, and/or cercariae from the molluscan intermediate host) are in principle accessible. However, infection levels in the intermediate hosts are, in general, low (some estimates suggest <6% of sampled snails infected)^{9,10} even in highly endemic populations with extremely high prevalence and intensities within the human hosts. We did attempt sampling of sympatric water bodies at the same time as the school-based sampling for this study, and we only found a single snail shedding schistosome cercariae. We felt that analysis of cercariae from a single snail from a single site would not add value to our current manuscript – particularly as they are likely to be clonal. We do, however, hope to inspire further research in this area encompassing both life stages.

3. The divergence between the local Lake Victoria *S. mansoni* population and the one more inland as a comparator, also have some bearing on 2. For example, *Biomphalaria* inland is not the same as those populations in the lake. It is likely that a reason why there is divergence could be due to local snail-parasite selection to *B. pfeifferi* versus *B. choanomphala*. This adds another level of uncertainty to the mechanisms by which the genetic diversity of these parasites are shaped. Greater consideration of the biology of transmission is need to balance out the emphasis on praziquantel selection pressure. It is a shame there is no further genetic sampling of *S. mansoni* populations inland. Is it not possible to look at other *S. mansoni* data to compare to bolster this as it is a weakness in my mind when there is an inference across a broader spatial level. Some reference to <https://parasitesandvectors.biomedcentral.com/articles/10.1186/s13071-014-0524-4> [parasitesandvectors.biomedcentral.com] might help better position their speculations.

Agreed and amended. As detailed, we have rephrased our text and focus considerably to accommodate other factors that may explain the observed effects, either independent of, or in addition to, the impact of praziquantel pressure over space and time here. Sampling from inland regions was limited by lack of available parasite material due to the overall lower prevalence of schistosomiasis in this district. However, we do, as suggested, also refer further to other *S. mansoni* data cited to support our revised focus.

4. I commend the authors on their bioinformatics and associated redaction of their sequence information. This adds clear value to the 'baseline' diversity. By inference, with such diversity in a small place, what expectations are there for the *S. mansoni* Lake Victoria populations as a whole?

This is indeed an interesting point, and one in which we have expanded upon within our revised text. We know from previous population genetic studies using microsatellites and/or cox-1 ITS barcoding¹¹, that there is evidence of extensive gene flow of both *S. mansoni* and *S. haematobium* all around the Lake Victoria populations as a whole, encompassing Uganda, Tanzania and Kenya. However, within country clear sub-structuring also occurs,

such as most notably within Uganda in relation to *S. mansoni* populations from school children within Lake Albert being distinct from those in Lake Victoria regions, the explanation for which in terms of parasite, host and/or environmental effect (including potential differential MDA histories) remains to be understood.

It is possible to make some speculations on the effective populations size given Fst measures have been made and it is possible to make some inferences on minimum estimates which would satisfy infinite population assumptions.

We have included estimates of effective population size in our revised text (Supplementary figure 6 and 7). We have considered at some length how best to estimate effective population sizes (N_e), and related statistics such as the effective number of breeders (N_b), particularly in the context of demonstrating the predicted impact of MDA on the N_e/N_b , as has been shown for other organisms such as HIV¹². In previous population genetic studies of schistosomes using microsatellites, we, and others, have used a number of currently available tools to estimate N_e , but these have tended to give inconsistent results that varied greatly between different techniques, so we have had little confidence in using these approaches. We suspect much of this inconsistency is due to a number of differences between schistosome genetics and that for model organisms and humans, such as the longevity of schistosomes, leading to very highly overlapping generations, clonal expansion in the snail vector and our lack of knowledge of basic genetic and life history traits such as mutation and recombination rates or the typical generation times. Most tools to estimate N_e/N_b from genomic data tend to have been developed for humans, and it may not be possible to reliably apply these for schistosomes.

Further, as demonstrated by Barbosa et al.¹³ and Steinauer et al.¹⁴ sample size (including the number of siblings) is important for reliable size N_e and N_b in schistosome populations. As discussed within our manuscript, the current inhibitory cost of genomic studies of schistosome, limits the sample sizes (and sibling assessment) available. We do agree that this is an area for future research and new valid tools and techniques for use on species such as schistosomes are certainly warranted, and in particular here could help elucidate how, for example, any differences in life-cycle generation and transmission dynamics around the lake could impact N_e/N_b .

I think it important to tone down the search for genetic resistance and concentrate more on the fascinating population genetics which is coming to light which is quite different to how we might have originally perceived it.

Agreed. Whilst we have retained material on the search for genetic resistance within our revised manuscript, particularly given that being the main hypothesis behind the original study design and sample collection to correspond with that published by Crellen et al.³, we have expanded our scope in regards to what this unique genomic dataset reveals about the population structure of schistosomes in general under such endemic situations and differential selective pressures.

References

1. Crellen, T. *et al.* Whole genome resequencing of the human parasite *Schistosoma*

- mansoni reveals population history and effects of selection. *Sci. Rep.* **6**, 1–13 (2016).
2. World Health Organization. *A road map for neglected tropical diseases 2021–2030*. https://www.who.int/neglected_diseases/Ending-the-neglect-to-attain-the-SDGs--NTD-Roadmap.pdf (2020).
3. Crellen, T. *et al.* Reduced Efficacy of Praziquantel Against *Schistosoma mansoni* Is Associated with Multiple Rounds of Mass Drug Administration. *Clin. Infect. Dis.* **63**, 1151–1159 (2016).
4. Faust, C. L. *et al.* Two-year longitudinal survey reveals high genetic diversity of *Schistosoma mansoni* with adult worms surviving praziquantel treatment at the start of mass drug administration in Uganda. *Parasit. Vectors* **12**, 607 (2019).
5. Webster, J. P., Gower, C. M. & Norton, A. J. Evolutionary concepts in predicting and evaluating the impact of mass chemotherapy schistosomiasis control programmes on parasites and their hosts. *Evol. Appl.* **1**, 66–83 (2008).
6. Melman, S. D. *et al.* Reduced Susceptibility to Praziquantel among Naturally Occurring Kenyan Isolates of *Schistosoma mansoni*. *PLoS Neglected Tropical Diseases* vol. 3 e504 (2009).
7. Le Clec'h, W. *et al.* Whole genome amplification and exome sequencing of archived schistosome miracidia. *Parasitology* **145**, 1739–1747 (2018).
8. Parker, M. *et al.* Border parasites: schistosomiasis control among Uganda's fisherfolk. *Journal of Eastern African Studies* **6**, 98–123 (2012).
9. Hamburger, J. *et al.* A polymerase chain reaction assay for detecting snails infected with bilharzia parasites (*Schistosoma mansoni*) from very early prepatency. *Am. J. Trop. Med. Hyg.* **59**, 872–876 (1998).
10. Hailegebriel, T., Nibret, E. & Munshea, A. Prevalence of *Schistosoma mansoni* and *S. haematobium* in Snail Intermediate Hosts in Africa: A Systematic Review and Meta-analysis. *J. Trop. Med.* **2020**, 8850840 (2020).
11. Gower, C. M. *et al.* Population genetic structure of *Schistosoma mansoni* and *Schistosoma haematobium* from across six sub-Saharan African countries: implications

for epidemiology, evolution and control. *Acta Trop.* **128**, 261–274 (2013).

12. _____ Liu, Y. & Mittler, J. E. Selection dramatically reduces effective population size in HIV-1 infection. *BMC Evol. Biol.* **8**, 133 (2008).

13. _____ Barbosa, L. M. *et al.* The effect of sample size on estimates of genetic differentiation and effective population size for *Schistosoma mansoni* populations. *Int. J. Parasitol.* **48**, 1149–1154 (2018).

14. _____ Steinauer, M. L. *et al.* Non-invasive sampling of schistosomes from humans requires correcting for family structure. *PLoS Negl. Trop. Dis.* **7**, e2456 (2013).

Reviewers' Comments:

Reviewer #1:

Remarks to the Author:

The authors have done an excellent job in revising the original submission. I am happy to now recommend publication without further revision.

Warwick Grant

Reviewer #2:

Remarks to the Author:

The authors have done an admirable job addressing my concerns and those of the other reviewers. Nice paper!